# DEPART: Hierarchical Multi-Agent System for Multi-Turn Interaction

## Abstract

Large Language Models (LLMs) excel at short-horizon tasks but struggle in complex, long-horizon scenarios involving multi-turn interactions, multi-step reasoning, and selective multi-modal perception. Two core challenges in these settings are effective long-term planning and mitigating cross-modal distraction. Our empirical analysis shows that single LLM agent exhibits steep performance drops as interaction steps increase, underscoring the limitations of monolithic approaches. To overcome these challenges, we propose **DEPART**, a hierarchical multi-agent framework that decomposes planning, action execution, and visual understanding into specialized agents. Through its **D**ivide, **E**valuate, **P**lan, **A**ct, **R**eflect, and **T**rack cycle, DEPART supports dynamic task decomposition, feedback-driven adaptation, and selective vision grounding to reduce cost and improve robustness. Building on this architecture, we introduce Hierarchical Interactive Multi-turn Policy Optimization (HIMPO), a two-round post-training strategy that alternately optimizes planner and executor with dense role-specific and sparse task-level rewards to encourage specialization and coordinated long-horizon reasoning. Across WebArena-Lite, VisualWebArena, and AlfWorld benchmarks, DEPART with HIMPO consistently outperforms strong single-agent and post-trained baselines.

## 1 Introduction

Large Language Models (LLMs) have shown remarkable capabilities across diverse domains. However, their strengths are most evident in single-turn, non-interactive tasks, such as math problem solving (Shao et al., 2024; Yu et al., 2025) and code generation (Wei et al., 2025a). Despite this progress, current LLMs struggle with complex, dynamic scenarios that demand multi-step decision-making across multi-turn interactions with diverse solution spaces (Yao et al., 2023). Conventional approaches to long-horizon tasks have primarily relied on single-agent architectures that directly map user queries to low-level actions (Gur et al., 2024; Qi et al., 2025; Wei et al., 2025b; Yang et al., 2024), overlooking the benefit of multi-step planning. While recent methods have introduced intermediate planning steps to improve performance (Rawat et al., 2025), assigning both strategic planning and precise execution to a single model introduces a bottleneck (Yang et al., 2025b).

To motivate the separation of planning and execution, we conduct a preliminary study using Claude 3.7 (Anthropic, 2025) in a single-agent setting on the shopping category of WebArena-Lite benchmark (Zhou et al., 2023a; Liu et al., 2025a). Using human-annotated oracle action counts as a proxy for task complexity (detailed in Appendix C.2), we observe that performance degrades as the number of required interaction steps increases in Figure 1. This trend suggests that long-horizon task failures may stem from the difficulty of jointly managing high-level planning and fine-grained execution in a single model (Erdogan et al., 2025).

In addition to separating planning and execution, we enforce a modular separation of vision and action, which enhances both efficiency and robustness by empowering the planner to dynamically determine when visual understanding is required. This design is motivated by two core observations. First, vision-language models are computationally expensive, and many steps in long-horizon tasks do not require visual input. Unnecessary

inclusion of visual context increases inference costs and may degrade performance due to cross-modal distraction (Shen et al., 2025), where irrelevant modality signals interfere with reasoning, as supported by our experiments in Appendix in Table 6. This challenge is not addressed by the text-only data generation pipelines in mulit-agent systems in prior works (Erdogan et al., 2025; Chen et al., 2025a). Second, training a unified multi-modal agent to handle both vision and action introduces fragility. Alignment fine-tuning can impair generalization (Zhai et al., 2024; Springer et al., 2025), and such agents require substantially more training data (Huang et al., 2025; Wan et al., 2025a). Moreover, reasoning gains achieved through reinforcement learning (RL) in LLMs do not seem to effectively transfer to large vision-language models (Wu et al., 2025c).

Concretely, we propose **DEPART**, a hierarchical multi-agent framework for long-horizon, multi-turn inter-action, illustrated in Figure 2 and detailed in Section 3. DEPART decomposes complex task into modular sub-problems with three specialized agents: a planner that generates sequential high-level plan steps; an action executor that performs grounded actions in the environment; and a vision executor that interprets visual context and shares relevant information. These agents operate within a structured communication loop that supports replanning and retry mechanisms (Bensal et al., 2025; Erdogan et al., 2025), enabling dynamic adaptation and error recovery. In contrast to prior works (Li et al., 2023; Hong et al., 2024; Wu et al., 2025d), our planner assigns only one plan step at a time, allowing the system to focus on the current interaction and dynamically adapt to environmental feedback before proceeding. **DEPART** stands for **D**ivide, **E**valuate, **P**lan, **A**ct, **R**eflect, and **T**rack, capturing the core stages of intelligent behavior, ensuring progress evaluation, reflection, and long-term coherence.

While DEPART's modular structure improves system-level coordination, LLMs pre-trained on general corpora often underperform in interactive, sequential environments. To address this, we introduce a post-training optimization framework, Hierarchical Interactive Multi-turn Policy Optimization (HIMPO), inspired by multi-agent post-training (Park et al., 2025; Wan et al., 2025b; Leong and Wu, 2025). HIMPO alternately trains the planner and executor across two rounds using the same optimization framework, critically differing in the reward design. The first round applies dense, role-specific feedback to foster strategy exploration (Yang et al., 2025b), support the decomposition of high-level goals into executable actions (Erdogan et al., 2025), and compensate for the lack of decision-relevant data in the pre-training of open-source LLMs (Yao et al., 2023; Wei et al., 2025b); the second round uses sparse, task-level rewards to align behavior with overall task success and mitigate reward hacking (Kumar et al., 2025). This curriculum encourages both specialization and coordinated long-horizon reasoning. Details are provided in Section 4.

We evaluate DEPART on WebArena-Lite (Zhou et al., 2023a; Liu et al., 2025a) and VisualWebArena (Koh et al., 2024a), two challenging web-based benchmarks, as well as AlfWorld in embodied environment for long-horizon interaction. Our experiments show that DEPART consistently outperforms strong single-agent baselines across various LLM backbones. Building on the DEPART architecture, we further boost performance using a smaller open-source model (Qwen3-4B) through two-round HIMPO., which enhances planner and executor via role-specific and task-level optimization.

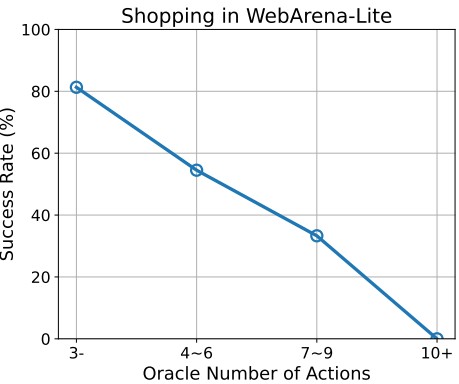

Figure 1: Success rate of single-agent (Claude 3.7) on shopping tasks in WebArena-Lite. X-axis represents the number of actions (i.e., interaction steps) required to complete each task. 3- denotes tasks solvable in 3 steps or fewer, while 10+ denotes tasks requiring more than 10 steps.

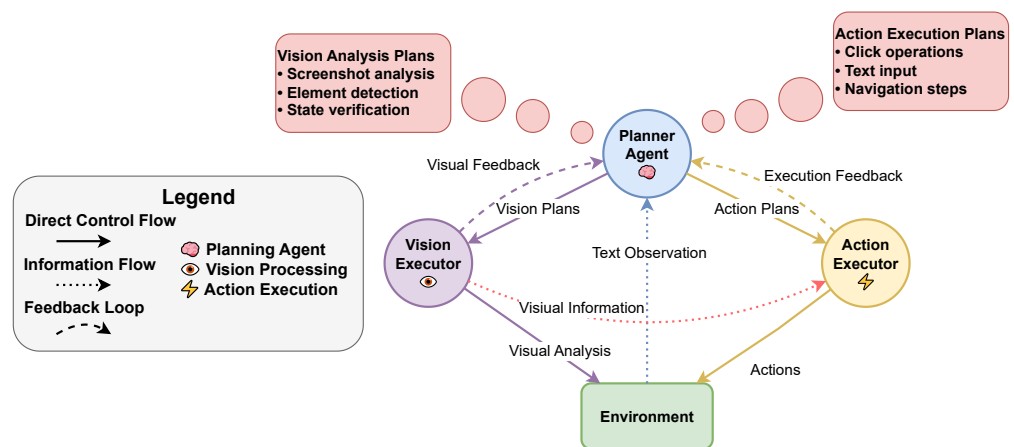

Figure 2: **Overview of DEPART**: The framework (1) divides tasks into distinct modalities and stages to reduce complexity; (2) evaluates the environment and task progress via observations and feedback; (3) plans high-level strategies based on evolving context; (4) acts through specialized executors for grounded interaction and vision understanding; (5) reflects on action outcomes to guide replanning; and (6) tracks global task status and historical context to maintain coherence across multiple turns.

## 2 RELATED WORK

Our work lies at the intersection of web-based agent systems and RL post-training for LLMs. We organize related work into two areas: (1) benchmarks and agent architectures for interactive web-based decision-making, and (2) RL methods for enhancing LLMs in long-horizon, multi-turn settings. Due to space constraints, we summarize key developments here and include full discussions in Appendix B. Recent advances in web agents have been driven by increasingly realistic and interactive benchmarks, progressing from synthetic environments like WoB (Shi et al., 2017) and MiniWoB++ (Liu et al., 2018) to high-fidelity platforms such as WebArena (Zhou et al., 2023a). These benchmarks have enabled the emergence of various agent architectures, including domain-specific models with lightweight policy heads (Furuta et al., 2024; Deng et al., 2024), prompt-based LLM agents leveraging modular tool-use strategies (Song et al., 2024a; Koh et al., 2024b), and RL-enhanced agents with post-training (Wei et al., 2025b; Qi et al., 2025). RL has shown promise in aligning LLMs with downstream objectives in single-turn scenarios (Shao et al., 2024; Yu et al., 2025; Liu et al., 2025b; Ouyang et al., 2022; Zhao et al., 2025b). Recent methods extend to multi-turn interactive tasks (Zhou et al., 2025; 2024b; Wei et al., 2025b; Qi et al., 2025) while most prior RL-based approaches rely on sparse, final-state rewards or make simplifying assumptions about the environment (Wei et al., 2025b; Qi et al., 2025), limiting their adaptability in complex, long-horizon tasks. In contrast, our method introduces fine-grained, agent-specific reward functions tailored to the planner and executor roles within a collaborative framework. We further propose a unified turn-level policy gradient objective that improves training stability while enabling expressive, role-specific behavior in multi-agent, sequential decision-making.

## 3 DEPART: HIERARCHICAL MULTI-AGENT SYSTEM

### 3.1 PROBLEM FORMULATION

A Markov Decision Process (MDP) is a standard framework for modeling sequential decision-making under full observability. A Partially Observable Markov Decision Process (POMDP) extends the MDP to settings

where the system follows MDP dynamics, but the agent only has partial access to the underlying state. We formulate a web browsing task as a POMDP to capture the inherent uncertainty and limited observability present in complex and dynamic web environments (Wei et al., 2025b). A POMDP is defined by the tuple $(\mathcal{S}, \mathcal{O}, \mathcal{A}, \mathcal{P}, \mathcal{R})$. The state space $\mathcal{S}$ encompasses the entire internet content, browser context, and user-related metadata. $\mathcal{O}$ denotes the observation space, which may consist of structured text $\mathcal{O}^{text}$ and image elements $\mathcal{O}^{img}$. As the full state $\mathcal{S}$ is prohibitively large and often not fully observable in practice, decision-making is based only on partial observations $\mathcal{O} = (\mathcal{O}^{text}, \mathcal{O}^{img})$. The action space $\mathcal{A}$ includes low-level browser interaction primitives such as clicking, typing, and scrolling. $\mathcal{P} : \mathcal{S} \times \mathcal{A} \to \Delta(\mathcal{S})$ is the transition probability ($\Delta(\mathcal{S})$ is the distribution over $\mathcal{S}$), capturing the environment dynamics given the current state and action. We denote $\mathcal{R} : \mathcal{S} \times \mathcal{A} \to \mathbb{R}$ as the reward function. Specifically, the agent receives a binary task reward $r_T \in \{0, 1\}$ at the final step $T$ of each task, indicating if task completion was successful (e.g., reaching a target page or submitting a correct form). Episodes terminate upon answer submission or after a fixed step limit. To facilitate learning, we further introduce dense, role-specific rewards in early training (e.g., execution alignment and plan confidence), along with penalties for format violations and step limits. In a second training stage, we remove these intermediate signals and optimize solely for task-level success. Full reward details are provided in Appendix D.4.

### 3.2 FROM SINGLE AGENT TO MULTI-AGENT

We begin by considering a single agent operating within the previously defined POMDP. The agent's policy $\pi_\theta$ is implemented on the top of an autoregressive language model, which serves as the decision-making backbone. To improve performance given a user query $\mathbf{x}$, the agent generates an explicit high-level multi-step plan $\mathbf{m}$, followed by a response $\mathbf{y}$ that encodes executable actions (Rawat et al., 2025). The plan $\mathbf{m}$ consists of reasoning and the corresponding plan.

During interaction with the web environment, the executed action at time step $t$, denoted as $a_t$, is abstracted from the response $\mathbf{y}_t$ using a deterministic mapping function $f$, i.e., $a_t = f(\mathbf{y}_t)$. For instance, if the response includes the span <act >do('Scroll Down') </act >, the resulting action is $a_t = $ do('Scroll Down'). The generation process for a single interaction step can be modeled probabilistically as:

$$\pi_\theta(\mathbf{y}, \mathbf{m}|\mathbf{x}, o) = \pi_\theta(\mathbf{y}|\mathbf{m}, \mathbf{x}, o) \cdot \pi_\theta(\mathbf{m}|\mathbf{x}, o) \tag{1}$$

where $\pi_\theta(\mathbf{y}, \mathbf{m}|\mathbf{x}, o)$ denotes the probability of producing a plan $\mathbf{m}$ and a response $\mathbf{y}$ given input query $\mathbf{x}$ and $o \in \mathcal{O}$. Assuming that $\mathbf{y}$ is conditionally independent of $\mathbf{x}$ given $\mathbf{m}$, the factorization simplifies to:

$$\pi_\theta(\mathbf{y}, \mathbf{m}|\mathbf{x}, o) = \pi_\theta(\mathbf{y}|\mathbf{m}, o) \cdot \pi_\theta(\mathbf{m}|\mathbf{x}, o) \tag{2}$$

This assumption reflects the intuition that once a coherent plan $\mathbf{m}$ is established, the query $\mathbf{x}$ provides no further information required for generating the response $\mathbf{y}$. However, this monolithic formulation demands that a single language model manages both planning and low-level execution simultaneously, limiting scalability and modularity.

To address these challenges, we build upon prior work (Erdogan et al., 2025; Wan et al., 2025b; Chen et al., 2025a) and propose a hierarchical multi-agent framework with three specialized agents with their individual policy: a planner $\pi_p$, a vision executor $\pi_v$, and an action executor $\pi_a$. Inspired by hierarchical planning in robotics (Zhang et al., 2025; Hsu et al., 2024a) and structured decision-making in MDPs (Pignatelli et al., 2024), the planner decomposes the user query $\mathbf{x}$ into a sequence of semantically meaningful plan steps (i.e., sub-goals), such as 'log in', 'navigate to product page', or 'apply filters', which serve as intermediate waypoints toward task completion, and then orchestrates the execution process. Each plan step functions as a localized objective, enhancing interpretability and reliability, while enabling executors to operate over shorter, context-specific horizons. By focusing on the current sub-goal rather than the entire task, each executor improves sample efficiency and behavioral robustness.

The planner produces two high-level plans $(\mathbf{m}^v, \mathbf{m}^a)$: $\mathbf{m}^v$ for the vision executor and $\mathbf{m}^a$ for the action executor. The vision executor handles perception by processing visual content from the web environment, while the action executor performs fine-grained browser actions. Formally, the decision process at a single time step is modeled as:

$$\mathbf{y} \sim \pi_a(\mathbf{y}|\mathbf{v}, \mathbf{m}^a, o^{text}) \cdot \pi_v(\mathbf{v}|\mathbf{m}^v, o^{img}) \cdot \pi_p((\mathbf{m}^a, \mathbf{m}^v)|\mathbf{x}, o^{text}) \tag{3}$$

where the planner $\pi_p(\mathbf{m}^a, \mathbf{m}^v|\mathbf{x}, o^{text})$ generates two distinct plans $\mathbf{m} = (\mathbf{m}^a, \mathbf{m}^v)$ for action and vision executors given query $\mathbf{x}$ and $o^{text}$. The vision executor then produces vision-derived information $\mathbf{v}$, and the action executor generates the response $\mathbf{y}$.

### 3.3 Multi-Turn Interactions

To handle long-horizon interactions, we extend this model over $T$ time steps:

$$\mathbf{y}_T \sim \prod_{t=1}^{T} \pi_a\left(\mathbf{y}_t|\mathbf{v}_t, \mathbf{m}_t^a, o_t^{text}\right) \cdot \pi_v\left(\mathbf{v}_t|\ \mathbf{m}_t^v, o_t^{img}\right) \cdot \pi_p\left((\mathbf{m}_t^a, \mathbf{m}_t^v)|\mathbf{x}, o_t^{text}, \{\mathbf{v}, \mathbf{m}^a, \mathbf{m}^v, \mathbf{y}\}_{<t}\right) \tag{4}$$

Execution unfolds in a sequential loop. Given a user query and the current text-based observation $o_t^{text} \in \mathcal{O}^{text}$, the planner generates two parallel but distinct high-level plans: one for the vision executor and one for the action executor. These plans consist of ordered directives tailored to the perception and interaction subroutines needed to complete the task. Importantly, the planner may omit visual planning steps entirely if it determines that visual input is unnecessary, thereby avoiding the overhead of vision processing when it does not contribute meaningfully to task completion.

During execution, the planner dispatches individual plan steps to the corresponding executor and tracks their progress internally. When visual perception is required, the vision executor processes the web page image $o_t^{img} \in \mathcal{O}^{img}$ and extracts relevant information, which is communicated to both the planner and the action executor. The action executor then integrates this visual feedback (when available), textual observations $o_t^{text} \in \mathcal{O}^{text}$, and planner directives to perform fine-grained interactions within the web environment.

The planner continuously monitors execution feedback and updated textual content to decide whether to: (*i*) proceed to the next plan step, (*ii*) retry the current step, or (*iii*) generate a new plan instead of purely update the pan at each step (Erdogan et al., 2025; Chen et al., 2025a). Throughout the process, the planner maintains an internal representation of executor progress and task state, enabling adaptive coordination and real-time plan revision. An overview of the full framework is illustrated in Figure 2.

## 4 Hierarchical Interactive Multi-turn Policy Optimization

To reduce data requirements, computational overhead, and instability associated with multi-modal post-training (Huang et al., 2025; Wan et al., 2025a; Zhai et al., 2024; Springer et al., 2025), we focus training on the high-level planner and low-level action executors. This leverages the observation that static image interpretation by the vision executor is generally less dependent on long-horizon context and can operate effectively without additional post-training.

We formulate this multi-agent system as a two-player Markov game (Shapley, 1953) between the planner and the action executor, treating the vision executor as part of the environment and excluding it from the training loop. Extending the standard POMDP framework, we define a Partially Observable Markov Game (POMG) by the tuple $(\mathcal{S}, \mathcal{O}, \mathcal{A}, \mathcal{M}, \mathcal{P}, \mathcal{R}^p, \mathcal{R}^a)$, where $\mathcal{S}, \mathcal{O}, \mathcal{A}$ retain their usual definitions in a POMDP. Specifically, $\mathcal{A}$ is the action space of the action executor and contains low-level browser interaction primitives. $\mathcal{M}$ is the space of plan messages from the planner, where $\mathbf{m}_t \in \mathcal{M}$ denotes the plan assigned to the action

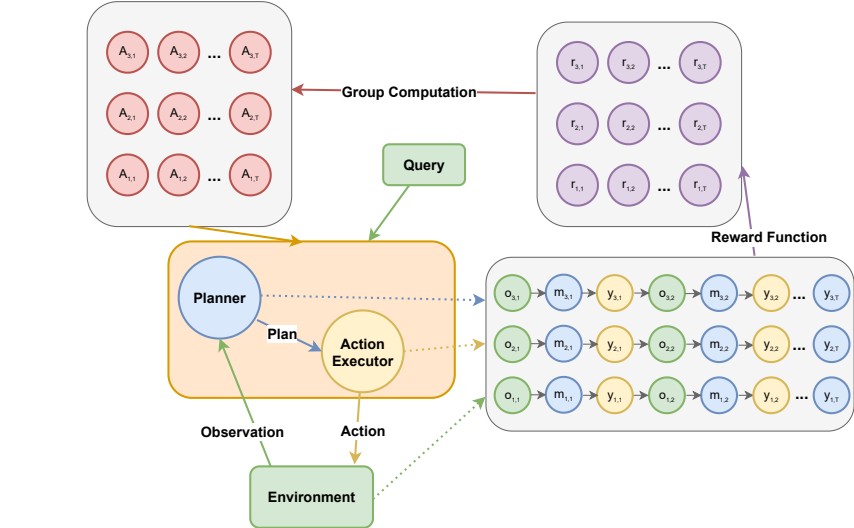

Figure 3: Hierarchical Interactive Multi-turn Policy Optimization (HIMPO). Parallel group rollouts generate trajectories for planner and action executor, with collective advantage computation for policy optimization. The vision executor is excluded from training.

executor at time step $t^1$ $\mathcal{P} : \mathcal{S} \times \mathcal{A} \times \mathcal{M} \to \Delta(\mathcal{S})$ is the transition probability. $\mathcal{R}^p, \mathcal{R}^a : \mathcal{S} \times \mathcal{A} \times \mathcal{M} \to \mathbb{R}$ are the reward functions for the planner and action executor, respectively, where the step-wise reward for the planner is built upon intrinsic reward (Kang et al., 2025; Zhao et al., 2025a) while the executor receives reward if the executor's actions align with the planner's expectations. This setup yields a general-sum game rather than a purely cooperative or zero-sum structure.

We adopt a parameter-sharing strategy in which the planner and the action executor share the same model weights $\theta$ but are distinguished by role-specific system prompts $S_p$ and $S_a$ (Wan et al., 2025b). This design avoids costly GPU model swaps during rollouts, supports larger batch sizes, and enables efficient joint optimization within a unified training pipeline. Building on Group Relative Policy Optimization (GRPO), we introduce Hierarchical Interactive Multi-turn Policy Optimization (HIMPO), a novel training framework that alternatively optimizes the planner and the action executor. In HIMPO, each agent has a distinct learning objective corresponding to its role $\in \{\text{plan, action}\}$:

$$
\mathcal{J}_{\text{HIMPO}}^{role}(\theta) = \mathbb{E}_{(\mathbf{x},\mathbf{e}) \sim \mathcal{D}, \{\tau_i\}_{i=1}^{G} \sim \pi_{\theta_{old}}(\mathcal{P}|(\mathbf{x},\mathbf{e}))} \left[ \frac{1}{\sum_{i=1}^{G} |\tau_i^{role}|} \sum_{i=1}^{G} \sum_{t=1}^{|\tau_i^{role}|} \right.
$$
$$
\left. \left( \min \left( l_{i,t}^{role}(\theta) A_{i,t}, \text{clip}(l_{i,t}^{role}(\theta), 1 - \epsilon, 1 + \epsilon) A_{i,t} \right) - \beta D_{\text{KL}}(\pi_\theta || \pi_{\text{ref}}) \right) \right] \tag{5}
$$

---

[1]We omit the superscript of $\mathbf{m}_t^a$ because we exclude vision executor during training.

$$l_{i,t}^{plan}(\theta) = \exp\left(\frac{1}{|\mathbf{m}_{i,t}|} \sum_{j=1}^{|\mathbf{m}_{i,t}|} \log \frac{\pi_\theta(\mathbf{m}_{i,t,j}|\mathbf{x}, \{\mathbf{m}_{i,\cdot}, \mathbf{y}_{i,\cdot}\}_{<t}, \mathbf{m}_{i,t,<j}, o_t^{text})}{\pi_{\theta_{old}}(\mathbf{m}_{i,t,j}|\mathbf{x}, \{\mathbf{m}_{i,\cdot}, \mathbf{y}_{i,\cdot}\}_{<t}, \mathbf{m}_{i,t,<j}, o_t^{text})}\right) \tag{6}$$

$$l_{i,t}^{action}(\theta) = \exp\left(\frac{1}{|\mathbf{y}_{i,t}|} \sum_{j=1}^{|\mathbf{y}_{i,t}|} \log \frac{\pi_\theta(\mathbf{y}_{i,t,j}|\mathbf{m}_{i,t}, \mathbf{y}_{i,t,<j}, o_t^{text})}{\pi_{\theta_{old}}(\mathbf{y}_{i,t,j}|\mathbf{m}_{i,t}, \mathbf{y}_{i,t,<j}, o_t^{text})}\right) \tag{7}$$

$$\hat{A}_{i,t} = \frac{r_{i,t} - \text{mean}(\{r_{i,t}\}_{i=1}^G)}{\text{std}(\{r_{i,t}\}_{i=1}^G)} \tag{8}$$

Our unified objective in equation 5 can be viewed as a multi-turn variant of dynamic sampling policy optimization (DAPO) (Yu et al., 2025) with turn-level policy gradient loss for long reasoning in 2-agent setting, illustrated in Figure 3. The key distinction between two agents are their trajectories $\tau_i^{role}$ and turn-level ratio $l_{i,t}^{role}$. For each task query $\mathbf{x}$, we sample a group of trajectories $\{\tau_1, \tau_2, .., \tau_G\}$, where $\tau_i^{plan}$ indicates a sequence of plan steps from the planner and $\tau_i^{action}$ denotes the sequence of low-level browser interaction primitives. The turn-level ratio for both agents are listed in equation 6 and equation 7. We extend the advantage computation in a group-relative manner to consider multi-turn steps $t$ as in equation 8.

## 5 EXPERIMENTS

We evaluate our approach in two stages: (1) measuring the impact of the DEPART architecture on long-horizon web tasks, and (2) analyzing the effectiveness of the HIMPO post-training algorithm via ablation studies. Experiments are conducted on complex web-browsing tasks (Appendix C.1) where agents use 12 high-level actions via Playwright (Koh et al., 2024a) and receive observations from the accessibility tree, a semantically rich DOM representation. Visual elements are linked to image assets when grounding is required. We report results on two web-agent benchmarks: WebArena-Lite (Zhou et al., 2023a; Liu et al., 2025a) (165 text-dominant tasks) and VisualWebArena (Koh et al., 2024a; Zhang et al., 2024) (114 vision-intensive tasks) with full experiment details are in Appendix E.1. We additionally evaluate our method on the AlfWorld benchmark (Shridhar et al., 2021), a text-based household task simulator that requires multi-step reasoning to follow high-level instructions and provides sparse, binary success signals. Following standard protocols, we use the **seen** split for in-distribution evaluation and the **unseen** split to assess out-of-distribution generalization.

### 5.1 DEPART: IMPACT OF MODULAR AGENT DESIGN

We compare three configurations of our architecture: (i) a single-agent model handling both planning and execution; (ii) a 2-agent setup separating planner and executor; and (iii) the full 3-agent DEPART system, which includes a vision executor dynamically invoked by the planner. Baselines include general-purpose LLMs, reasoning-specialized models, and RL-fine-tuned agents. In web browsing tasks, to ensure fair comparison with prior work (Wei et al., 2025b; Qi et al., 2025), we first evaluate models without visual input on WebArena-Lite. As shown in Table 1, prompting strong foundation models (e.g., OpenAI-o3, OpenAI-o4-mini, Claude 3.7) yields over 35% success. RL post-trained agents like WebRL and WebAgent-R1 further improve to over 40% (Table 2). Notably, introducing modularity leads to consistent gains. For example, Claude 3.7 in the 2-agent configuration reaches 46.1% average success in Table 1, surpassing both WebRL and WebAgent-R1 in Table 2. These results highlight the benefit of structured collaboration between specialized agents. In AlfWorld in Table 3, we empirically confirm that modularity increases success rate in both Qwen3-4B and Claude 3.7.

Table 1: Task success rate in WebArena-Lite (Zhou et al., 2023a) for **prompting-only methods without visual input**. The highest value in each column is in bold. Avg SR denotes the average success rate.

| Category | Model | Reddit | GitLab | CMS | Map | Shopping | Avg SR |
|---|---|---|---|---|---|---|---|
| **1 Agent** | Qwen2.5-3B | 5.3 | 13.3 | 5.7 | 0.0 | 4.4 | 6.1 |
| | Llama3.1-8B | 5.3 | 10.0 | 5.7 | 15.4 | 8.9 | 8.5 |
| | Qwen2.5-32B | 10.5 | 20.0 | 20.0 | 19.2 | 17.8 | 16.9 |
| | GPT-4o | 10.5 | 10.0 | 20.0 | 20.0 | 11.1 | 13.9 |
| | GPT-4o-Turbo | 10.5 | 16.7 | 14.3 | 36.7 | 13.3 | 17.6 |
| | QwQ-32B | 15.8 | 33.3 | 25.7 | 15.4 | 20.0 | 22.4 |
| | OpenAI-o3 | 36.8 | 46.7 | **45.7** | **38.5** | 33.3 | 39.4 |
| | OpenAI-o4-mini | **47.4** | 43.3 | **45.7** | 26.9 | 28.9 | 36.9 |
| | Qwen3-4B | 5.3 | 0.0 | 11.4 | 8.3 | 26.7 | 12.1 |
| | Claude 3.7 | 42.1 | 16.7 | 40.0 | 16.7 | 57.8 | 35.8 |
| **2 Agents (ours)** | Qwen3-4B | 5.3 | 3.3 | 14.3 | 8.3 | 31.1 | 14.5 |
| | Claude 3.7 | **47.4** | **56.7** | 42.9 | 16.7 | **60.0** | **46.1** |

Table 2: Task success rate in WebArena-Lite (Zhou et al., 2023a) for **RL post-training methods without visual input**. The highest value in each column is in bold. Avg SR denotes the average success rate.

| Category | Model | Reddit | GitLab | CMS | Map | Shopping | Avg SR |
|---|---|---|---|---|---|---|---|
| **1 Agent** | Llama3.1-8B (WebRL) (Qi et al., 2025) | **63.2** | 46.7 | 54.3 | **36.7** | 31.1 | 42.4 |
| | Llama3.1-8B (WebAgent-R1) (Wei et al., 2025b) | 47.4 | **56.7** | 57.1 | 23.1 | 44.4 | 44.8 |
| | Qwen2.5-3B (WebAgent-R1) (Wei et al., 2025b) | 26.3 | 53.3 | 48.6 | 26.9 | 24.4 | 33.9 |
| | Qwen3-4B (MT-GRPO) | 26.3 | 43.3 | 54.3 | 16.7 | 42.2 | 37.6 |
| **2 Agents** | Qwen3-4B (HIMPO) (ours) | 52.6 | **56.7** | **62.9** | 27.8 | **57.8** | **51.5** |

## 5.2 HIMPO: Improving of Post-Training and Ablation Study

Beyond architectural improvements, we evaluate our proposed post-training algorithm, HIMPO. Following prior work (Wei et al., 2025b; Qi et al., 2025), we train on 647 WebArena tasks not included in WebArena-Lite, reserving the 165 human-verified WebArena-Lite tasks for evaluation. Table 2 reports results from prior methods and our own experiments. WebRL and WebAgent-R1 are state-of-the-art RL post-training methods on this benchmark, motivating us to explore whether our HIMPO can improve the performance of smaller models compared with those proprietary LLMs. Specifically, we use Qwen3-4B and include a single-agent baseline trained with multi-turn GRPO (denoted MT-GRPO), a direct extension of GRPO to multi-turn settings (Wei et al., 2025b). We show that applying HIMPO in a 2-agent configuration with separate planner and executor with two-round HIMPO post-training (bottom row in Table 2) consistently outperform all single-agent post-training, especially that it improves the success rate against MT-GRPO with the same model size (Qwen3-4B) in every category by a wide margin. We follow the same training pipeline for AlfWorld, showing that post-training Qwen3-4B with HIMPO outperforms single-agent post-training methods.

## 5.3 Ablation Studies

**RL post-training** To assess the contribution of individual components in HIMPO, we conduct ablations targeting four key design choices: (i) dense role-specific rewards, (ii) turn-level ratio estimation, (iii) dynamic sampling, and (iv) modular training. Methodological details are provided in Appendix E.2. We evaluate eight training variants, with M1–M4 trained in a single-agent setting, and report their success rates on WebArena-Lite across epochs (Figure 4).[2] Notably, all methods outperform the pre-trained base model (12.1%

---

[2]WebArena-Lite tasks are not used for training.

Table 3: Task success rate in AlfWorld (Shridhar et al., 2021) for **prompting-only methods**.

| Category | Model | Seen | Unseen |
|----------|-------|------|--------|
| | OpenAI-o3-mini | 58.9 | 62.7 |
| | Deepseek-R1 | 61.4 | 53.7 |
| | Qwen2.5-7B-Instruct | 72.1 | 76.1 |
| **1 Agent** | Llama-3.1-8B-Instruct | 32.9 | 40.3 |
| | GPT-4 | 64.8 | 64.4 |
| | Qwen3-4B | 36.4 | 37.3 |
| | Claude 3.7 | 78.6 | 82.1 |
| **2 Agents** | Qwen3-4B | 42.1 | 41.0 |
| **(ours)** | Claude 3.7 | **82.1** | **85.1** |

Table 4: Task success rate in AlfWorld (Shridhar et al., 2021) for **post-training methods**.

| Category | Model | Seen | Unseen |
|----------|-------|------|--------|
| | Llama3.1-8B-Instruct (ETO) (Song et al., 2024b) | 78.6 | 71.6 |
| | Llama3.1-8B-Instruct (MPO) (Xiong et al., 2025) | 82.9 | 78.4 |
| | Llama3.1-8B-Instruct (BPO) (Wang et al., 2025a) | 87.9 | 89.6 |
| **1 Agent** | Qwen2.5-7B-Instruct (ETO) (Song et al., 2024b) | 82.1 | 76.1 |
| | Qwen2.5-7B–Instruct (MPO) (Xiong et al., 2025) | 81.4 | 88.1 |
| | Qwen2.5-7B–Instruct (BPO) (Wang et al., 2025a) | 90.0 | 91.0 |
| | Qwen3-4B (MT-GRPO) | 82.1 | 88.1 |
| **2 Agents** | Qwen3-4B (HIMPO) (ours) | **92.9** | **91.8** |

in Table 1) without supervised fine-tuning (SFT), indicating that tool-augmented agents can benefit from RL alone. Building on M1, M2 introduces dense, role-specific rewards (details in Appendix D.4) and M3 apply turn-level ratio clipping (Wan et al., 2025b), by treating an entire agent turn as a single action. Both contribute to further performance gains by better aligning reward scaling with the underlying MDP structure compared with token-level ratios (Shao et al., 2024; Yu et al., 2025).

To isolate the effect of modular training, M5 trains only the planner, and M6 trains both planner and executor. The progression M3 → M5 → M6 shows consistent gains, confirming that combining planning and execution in a single model can create a performance bottleneck (Yang et al., 2025b). Although M6 (single-round HIMPO) already outperforms all single-agent baselines, early training is slower, likely due to multi-agent optimization complexity and susceptibility to reward hacking from output format violations (Figure 4). To better understand this issue, a preliminary study with dynamic sampling (Yu et al., 2025) (M4) over-samples and filters trajectories with incorrect output formats, accelerating early-stage convergence but plateauing sooner, highlighting a trade-off between early stability and long-term generalization. This insight motivates our two-round HIMPO strategy: M7 uses the first 6 epochs as a warm start with role-specific rewards, format penalties, and dynamic sampling, and M8 continues training using only task-level rewards. M8 consistently improves over M6, demonstrating the benefit of curriculum-style refinement (Qi et al., 2025) in reward design.

**Cross-modal Distraction**  We also study the effect of adding visual input. In WebArena-Lite, injecting vision often introduces irrelevant context or reasoning overhead, confirming the cross-modal distraction phenomenon (Shen et al., 2025) (green vs. red cells in Appendix in Table 6; detailed cases in Appendix G). Conversely, on VisualWebArena, all models benefit from visual input when tasks require perception (Table 7 in Appendix E.3). Motivated by this contrast, our final configuration adopts the full 3-agent DEPART framework, in which the planner selectively invokes visual processing only when needed. Because Qwen3-4B lacks

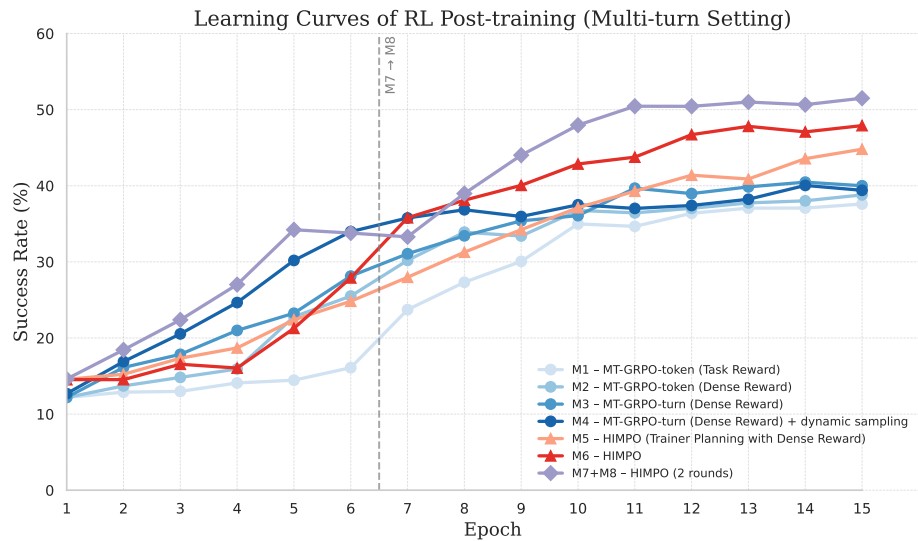

Figure 4: Evaluation on WebArena-Lite across training epochs. Methods M1–M4 correspond to single-agent training approaches, whereas M5–M8 are multi-agent training approaches. The highest success rate is achieved by initializing with M7 (warm start) and continuing training with M8; the dashed line marks the point of this transition.

native vision capability, we employ Claude 3.7 as its vision executor within the three-agent setup, combined with two-round HIMPO training (M7 + M8). This approach yields a 52.1% success rate on WebArena-Lite, surpassing all baselines and demonstrating that selectively grounding vision improves performance. Moreover, applying the three-agent configuration with HIMPO on Qwen3-4B raises its success rate on VisualWebArena tasks from 9.6% to 36.0% (Table 7), further highlighting the benefit of targeted visual grounding.

## 6 CONCLUSION

This work addresses key limitations of LLM-based agents in solving complex, long-horizon tasks. We propose DEPART, a modular multi-agent framework that separates planning, vision, and execution into specialized components, enabling structured coordination through dynamic communication, replanning, and retries. To complement this architecture, we introduce HIMPO, a two-stage post-training algorithm that improves learning efficiency through role-specific and task-level optimization. Together, DEPART and HIMPO significantly improve performance on realistic web-based and embodied benchmarks. We view structured multi-agent systems as a necessary foundation for advancing post-training and long-horizon decision-making in LLM-based agents.

## 7 ETHICS STATEMENT

This research adheres to the ICLR Code of Ethics. All datasets used are publicly available and open-source, with licenses permitting research use. Human annotation was performed solely to estimate oracle action counts for evaluation purposes, based on known task outcomes in existing benchmarks. No private or personally identifiable information was involved, and no new data from human subjects was collected. The study does not raise privacy, security, or fairness concerns, and we disclose no conflicts of interest or external sponsorships.

## 8 REPRODUCIBILITY STATEMENT

The main training equations are presented in Section 4, and the algorithmic designs are detailed in Appendix D. All datasets used in our experiments are publicly available and described in Section 5 and Appendix E.1. Implementation details, computational resources, and system prompts are also provided in Appendix F. Taken together, these resources enable independent researchers to verify and reproduce our findings.

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

## A    LLM Usage Disclosure

We used Large Language Models (LLMs), specifically OpenAI's ChatGPT, to aid in writing and editing. The LLM assisted with grammar, phrasing, and improving clarity, but did not contribute to research ideation, experimental design, analysis, or result interpretation. All content was authored and verified by the human authors.

## B    Full Related Work

### B.1    Web Benchmarks and Agents

The development of web-browsing agents has progressed hand-in-hand with the creation of benchmarks designed to evaluate their reasoning and interaction capabilities. Early efforts, such as WoB (Shi et al., 2017) and MiniWoB++ (Liu et al., 2018), introduced synthetic environments composed of simple website widgets. These benchmarks supported basic mouse and keyboard interactions but lacked the complexity and variability required to reflect real-world web tasks.

To more closely approximate practical scenarios, WebShop (Yao et al., 2022) simulated a large-scale e-commerce platform containing rich product data and goal-directed user instructions. While offering more realistic content than earlier benchmarks, its scope was confined to a single domain with limited interactivity, constraining its utility for evaluating general-purpose web agents. As LLMs have become increasingly capable of processing structured web data such as HTML and DOM trees (Tan et al., 2025), newer benchmarks have sought broader coverage and greater realism. For example, Mind2Web (Deng et al., 2023) extended the task set to diverse websites using human-recorded demonstrations. Nevertheless, its static nature and limited interactivity restrict its ability to evaluate adaptive behavior.

More recent benchmarks have emphasized interactivity reproducibility, and higher fidelity. WebWalk-erQA (Wu et al., 2025b) evaluates text-based reasoning abilities in a question–answer format while constraining actions to simple 'click' operations, thereby focusing on navigation and information-seeking capabilities. BrowseComp-VL (Geng et al., 2025) further increases task complexity by requiring agents to perform multi-step information retrieval involving both visual and textual inputs. WebArena (Zhou et al., 2023a) introduced a suite of simulated websites with real functionality, enabling realistic and controlled experimentation and quickly becoming a central benchmark for testing LLM-based web agents in dynamic settings. VisualWebArena (Koh et al., 2024a) extended WebArena by integrating visual understanding with language processing.

Building on these benchmarks, a diverse array of web agents has emerged (Nakano et al., 2022; Wei et al., 2025b; Wu et al., 2025b; Qi et al., 2025; Zhou et al., 2023b; 2024a). Broadly, these agents can be grouped into three categories. First, domain-specific agents typically rely on smaller LLMs trained or fine-tuned to select relevant HTML elements or execute low-level actions (Furuta et al., 2024; Deng et al., 2024). Second, prompt-based agents leverage large foundation models orchestrated through prompting strategies or modular tool-use workflows to tackle complex navigation tasks (Song et al., 2024a; Koh et al., 2024b; He et al., 2024).

Our work aligns with a third line of research: post-training methods for improving web agents. Prior approaches in this direction typically reduced web scenarios to question–answer pairs (Wu et al., 2025a; Li et al., 2025; Geng et al., 2025) or relied on sparse, trajectory-level rewards provided only upon task completion (Qi et al., 2025; Wei et al., 2025b). Such reward structures posed significant challenges for credit assignment and efficient exploration, particularly in long-horizon settings. For instance, WebRL (Qi et al., 2025) trained a reward model that judges success solely based on the final state of the webpage, assuming that reaching the correct page equates to task completion. In practice, however, agents may navigate to the correct page yet fail to extract or generate the required output. WebAgent-R1 (Wei et al., 2025b) performed clipping at the token level and aggregates the resulting loss directly, introducing bias and leads to training instability over long sequences. In addition to these algorithmic limitations, many RL-based approaches assumed simplified

environments, overlooking the challenges inherent in multi-turn interactions and multimodal observation handling in real-world web settings (Qi et al., 2025; Wei et al., 2025b; Wu et al., 2025a).

Our work addresses these gaps by incorporating fine-grained, agent-specific reward functions tailored to sub-goal completion and action relevance within a multi-agent setting. We further extend agent capabilities to handle multi-turn interactions and visual inputs, enabling robust behavior in visually complex and dynamic web environments (Abuelsaad et al., 2024; Iong et al., 2024). Moreover, we propose a turn-level training objective designed for stable optimization in hierarchical multi-agent settings, mitigating the instability and credit-assignment issues observed in prior methods. Other efforts in the multi-agent setting for web tasks (Erdogan et al., 2025) have proposed synthetic data generation pipelines for fine-tuning. While orthogonal to our focus, these methods could be conceptually integrated with our approach in future work.

## B.2 REINFORCEMENT LEARNING FOR LLMs

Reinforcement learning has proven effective for aligning LLMs with downstream objectives, but most existing approaches have focused on single-turn settings (Shao et al., 2024; Casper et al., 2024; Ouyang et al., 2022; Ziegler et al., 2022; Christiano et al., 2017). Algorithms such as PPO (Ouyang et al., 2022; Ramamurthy et al., 2023), GRPO (Shao et al., 2024), and DPO (Rafailov et al., 2023) became standard in this regime. In contrast, multi-turn long-horizon applications, where an agent must reason, plan and act through sequential interactions, pose optimization difficulties that remain insufficiently addressed in domains such as web navigation (Zhou et al., 2024b; Wei et al., 2025b), embodied planning (Fei et al., 2025), and multi-turn mathematical reasoning (Wan et al., 2025b; Zheng et al., 2025).

Many studies (Chen et al., 2025b; Jin et al., 2025; Feng et al., 2025a) framed multi-turn tasks as bandit problems, relying solely on outcome-level rewards such as answer or format correctness. This formulation was inadequate for long-horizon reasoning because it treated the entire trajectory as a single decision step and ignored turn-level signals indicating whether intermediate steps were helpful or harmful. A natural remedy was to learn a process-level reward model or critic model, but this typically required expensive on-policy data collection and did not generalize well with limited fine-tuning data (Setlur et al., 2025; Zhou et al., 2024b; 2025).

Another line of work attempted to directly adapt successful single-turn algorithms to multi-turn objectives (Wei et al., 2025b; Qi et al., 2025; Wang et al., 2025b). RAGEN (Wang et al., 2025b), for instance, concatenated all states, intermediate reasoning, and actions into a unified episode-level response, which created scalability challenges in long-horizon tasks. WebAgent-R1 and WebRL also lacked explicit turn-level credit assignment (Wei et al., 2025b; Qi et al., 2025). To address this, GiGPO (Feng et al., 2025b) introduced a two-level structure for estimating relative advantage with an additional anchor-state grouping mechanism, but this approach struggled in highly complex environments where identical states were hard to detect due to noise or subtle differences. Zeng et al. (2025) incorporated a turn-level advantage estimation strategy to enable more precise credit assignment in multi-turn agent interactions, but it was evaluated only in two-turn tool-use settings. IPO (Fei et al., 2025) modeled turn-level optimization by treating different tokens in GRPO as distinct decision steps. Although this better reflected multi-turn structure, the cumulative product of token-level ratios shrunk rapidly as sequence length grew, hitting the clipping threshold early and introducing exponentially worsening bias.

Our proposed algorithm, HIMPO, leverages a turn-level ratio combined with token-level averaging strategies (Wan et al., 2025b; Zheng et al., 2025) and a two-round post-training scheme. In the first round, we introduce fine-grained reward function to encourage exploration, and in the second round, we apply standard task rewards to promote improved generalization.

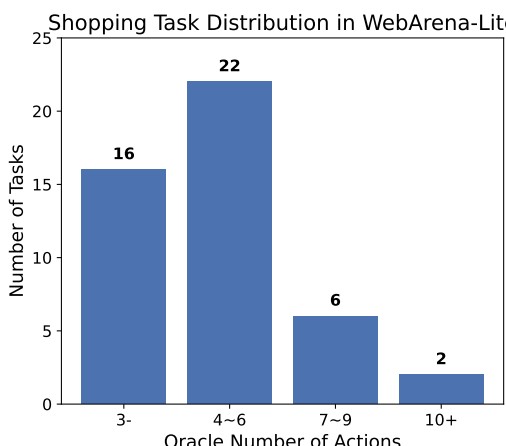

Figure 5: Distribution of shopping tasks in WebArena-Lite, categorized by the oracle number of required actions. The dataset includes 46 tasks spanning a range of interaction complexities.

## C  PROPERTIES OF WEB BROWSING TASKS

### C.1  CHALLENGE OF MULTI-TURN WEB BROWSING TASKS

In Figure 1, we empirically show that web browsing tasks become increasingly challenging as the number of interactions and action executions grows. This difficulty can also be understood through the lens of MDPs, particularly the concept of MDP depth (Pignatelli et al., 2024). Following (Ni et al., 2023), we define the depth of an MDP as the number of temporal steps that intervene between **a key action** and its observable outcome. In long-horizon web tasks, agents often encounter bottleneck decisions, i.e., critical actions that are required for eventual success but whose effects manifest only after a long delay. For example, in an e-commerce setting, clicking the 'log in' button and entering credentials is **a key action**: it may not yield any immediate reward but is necessary for accessing the order history page or editing account information later in the task. Despite the existence of many viable action paths, all successful trajectories must pass through such bottlenecks. When these decisions are temporally distant from their consequences, the MDP becomes deep, making it harder for the agent to maintain causal coherence and long-term focus through the interaction. We address this challenge by deriving sub-goal decomposition in Section 3.2 and extending it to multi-turn interactions in Section 3.3. This decomposition exposes bottleneck decisions as explicit intermediate objectives, reducing MDP depth and improving execution reliability by anchoring the agent's reasoning to structurally important waypoints.

### C.2  ANNOTATORS & ANNOTATION PROTOCOL

To support the observation that task difficulty increases with the number of required actions for an LLM agent (as illustrated in Figure 1), we perform a detailed analysis of the shopping category in WebArena-Lite (Zhou et al., 2023a; Liu et al., 2025a), which includes 46 diverse tasks. These tasks fall into three broad categories: webpage navigation, question answering, and content modification. We define the oracle action count as the minimal number of environment interactions needed to successfully complete a task. These oracle counts, grouped into discrete complexity ranges, are shown in Figure 5.

Among the three task types, webpage navigation is the most challenging to define optimally. This is due to the presence of multiple valid search strategies: basic keyword search, advanced search with filters, or navigating through category hierarchies. Since success in these tasks is judged by reaching the correct final

---

**Algorithm 1** Hierarchical Interactive Multi-turn Policy Optimization (HIMPO)

---

**Input:** $\pi_\theta$, Dataset $\mathcal{D}$, $\epsilon$, learning rate for planner and action executor $\alpha_p$, $\alpha_a$, batch size $B$.
**Output:** Parameter for the agent policy $\theta$.
1: Initialize planner $\pi_p = \pi_\theta(\cdot|S_p)$, action executor $\pi_a = \pi_\theta(\cdot|S_a)$ with system prompt $S_p$ and $S_a$
2: **for** each episode **do**
3:    **for** $i = 1 : B$ **do**
4:       $(\mathbf{x}, \mathbf{e}) \sim \mathcal{D}$
5:       collect group of trajectories in equation 4
6:    **end for**
7:    estimate advantage in equation 8
8:    calculate $\mathcal{J}_{\text{HIMPO}}^{plan}(\theta)$ in equation 5
9:    $\theta \leftarrow \theta + \alpha_p \nabla_\theta \mathcal{J}_{\text{HIMPO}}^{plan}(\theta)$
10:    **for** $i = 1 : B$ **do**
11:       $(\mathbf{x}, \mathbf{e}) \sim \mathcal{D}$
12:       collect group of trajectories in equation 4
13:    **end for**
14:    estimate advantage in equation 8
15:    calculate $\mathcal{J}_{\text{HIMPO}}^{action}(\theta)$ in equation 5
16:    $\theta \leftarrow \theta + \alpha_a \nabla_\theta \mathcal{J}_{\text{HIMPO}}^{action}(\theta)$
17: **end for**

---

URL, identifying the optimal path requires knowledge of that target in advance. Annotators leveraged the known final URL from (Zhou et al., 2023a; Liu et al., 2025a) to retrospectively trace the shortest path back to the start, yielding a reliable estimate of the oracle trajectory.

Despite the fact that LLM agents are evaluated solely on task success rather than trajectory optimality, we observe that the choice of search strategy implicitly affects success rate. In 7 shopping tasks with oracle action counts in the 4–6 range, the optimal strategy is advanced search or category navigation. However, LLM agents often defaults to basic keyword search, which introduces ambiguity. In cases where advanced search is optimal, keyword search typically leads to longer horizons, requiring inspection of multiple irrelevant pages. In other cases, the keyword search fails altogether if the query does not match any product title, whereas structured category navigation would succeed. For these 7 tasks, the success rate drop from 54.5% (across all tasks in the 4–6 oracle range) to 28.6% when isolating just those requiring more deliberate search strategies.

In contrast, question answering and content modification tasks are more straightforward for annotators. After logging in (if required), most steps involved clicking buttons on the same page or conducting linear navigation, such as browsing through paginated content (e.g., reviewing order history). These task types generally pose fewer ambiguities and show less performance degradation.

## D   ALGORITHM DESIGN

Algorithm 1 summarizes the practical implementation of HIMPO, which optimizing both planner and action executor agents using the alternating procedure until convergence. In each phase, a query $\mathbf{x}$ and the corresponding environment $\mathbf{e}$ are sampled from dataset $\mathcal{D}$. After collecting enough trajectories, we compute advantage values via equation 8 for updating parameters $\theta$ with different learning rate, inspired by two-time scale analysis (Das et al., 2024). Note that both agents share the same parameter $\theta$ in practice, so the entire training process can be viewed as a multi-objective optimization, where planner and action executor have their own objectives with the same parameters $\theta$.

### D.1 MITIGATING SFT WITH RELIABLE TOOLS

Supervised fine-tuning (SFT) is commonly used as a warm start for RL post-training, helping models memorize task-specific rules. However, SFT has well-known limitations, including poor generalization to out-of-distribution (OOD) scenarios (Chu et al., 2025) and susceptibility to catastrophic forgetting (Shenfeld et al., 2025). We observe that in structured tasks that can be decomposed hierarchically, the reliance on SFT can be partially mitigated through the use of reliable external tools. For instance, tools like Playwright for web browsing (Zhou et al., 2023a) or PID controllers for drone navigation (Hsu et al., 2024b) can handle low-level execution reliably, reducing the need for the model to memorize procedural rules via SFT. Moreover, SFT demands large-scale, high-quality training data, which can be difficult to obtain and may not align with the deployment distribution. By leveraging stable tool use, we reduce this dependency and enable more robust and generalizable learning.

### D.2 TWO-ROUND CURRICULUM: DROPPING AND REINTRODUCING KL DIVERGENCE

We adopt a two-round curriculum using HIMPO. In the first round, we remove the KL divergence term typically used in RL post-training. KL divergence acts as a regularizer by penalizing deviation from a reference policy, which is usually a SFT model (Ouyang et al., 2022; Shao et al., 2024). However, since we do not perform SFT prior to first-round RL, the only available reference would be the raw base model. Constraining the policy to stay close to this untuned model provides little practical benefit and may hinder effective learning. Removing the KL term not only simplifies training by eliminating the need to tune its coefficient but also reduces memory and computational overhead. More importantly, it enables more flexible exploration (Wan et al., 2025b), which is crucial in the early stages of learning. From a methodological perspective, general-purpose LLMs are not inherently optimized for dynamic or multi-turn reasoning (Feng et al., 2025c); thus, pattern drift during RL is both expected and beneficial. Removing the KL constraint allows the policy to adapt more freely to such interactions. In the second round of training, we reintroduce the KL divergence term to encourage the policy to remain closer to the improved reference policy obtained from the first round, thereby stabilizing further optimization.

### D.3 TURN-LEVEL RATIO AND POLICY GRADIENT LOSS

To unify our objectives under multi-turn interaction settings, we define a turn-level ratio based on sequence likelihood (Zheng et al., 2023; 2025), as shown in Equations equation 6 and equation 7. Unlike token-level formulations, the turn-level ratio aligns more closely with the underlying MDP structure by treating all tokens within a turn as a single action. This enables us to apply clipping at the turn level (Wan et al., 2025b), which improves stability and consistency in policy optimization.

Building on this formulation, we further compute the policy gradient loss at the turn level. Specifically, the loss is averaged over the total number of turns across all sampled trajectories, representing a multi-turn variant of DAPO (Yu et al., 2025). This design helps address two key limitations commonly observed in previous multi-turn methods (Fei et al., 2025; Wei et al., 2025b; Wan et al., 2025b).

First, in long trajectories consisting of multiple high-quality steps, prior approaches often dilute the learning signal across many individual actions, weakening the model's ability to capture reasoning-relevant patterns. Second, excessively long turns, which often contains low-quality content such as repetition or irrelevant actions, tend to be under-penalized. Our turn-level formulation mitigates both issues by emphasizing coherent turn-level decision-making and discouraging verbose, low-quality outputs.

### D.4 REWARD DESIGN

Our Hierarchical Multi-Agent Intrinsic Policy Optimization (HIMPO) is trained in two stages, each designed to balance learning stability and generalization for multi-turn tasks that benefit from dividing labor between strategic planning and concrete action execution. In both rounds, agents optimize task success via reinforcement learning, but we structure rewards differently to facilitate learning.

**Two-Round Training Procedure**

- First Round (Dense + Format-Aware Rewards): We employ dense, role-specific rewards for both planner and executor, supplemented with penalties for incorrect output $r_t^p = -0.5$ formatting and exceeding episode limit $r_t^p = -0.1$. This setup promotes stable credit assignment and reduces noise during early training. We also optionally use dynamic sampling (Yu et al., 2025) to over-sample and filter out outputs with format errors, ensuring that policy updates are based on well-formed and executable actions.

- Second Round (Sparse Final Reward Only): Once the agents have developed stable behavior, we remove all intermediate and format-related signals and train using only the final binary task reward ($r_T \in 0, 1$), defined in Section 3.1. This second round encourages agents to generalize beyond hand-crafted signals and optimize purely for end-task success.

The following subsections detail our fine-grained reward design for the first round, which enables efficient learning within the hierarchy.

### D.4.1 REWARD FOR THE LOW-LEVEL EXECUTOR

Separating the planner from the action executor in a hierarchical framework naturally creates sub-goals for the executor as the high-level query is decomposed into a sequence of planning steps. This setup allows us to provide a step-wise reward measuring how closely the executor's actions align with the planner's expectations. Specifically, we define the step-wise reward for the action executor as $r_t^s = 1/T_{max}$, where $T_{max}$ is the predefined time step limit.

### D.4.2 REWARD FOR THE HIGH-LEVEL PLANNER

Designing a dense reward for the planner is more challenging because its plans do not immediately yield observable environmental outcomes. We therefore introduce an intrinsic reward (Kang et al., 2025; Zhao et al., 2025a) based on the planner's own confidence in its generated plans. The intuition is to incentivize the planner to produce decisive and unambiguous plans, which are more likely to be coherent and executable by the low-level action executor.

For plan step $\mathbf{m}_t$[3], the intrinsic reward is defined as the average confidence score across its $N$ tokens:

$$r_t^s(\mathbf{m}_t|o_t^{text}) = \frac{1}{N} \sum_{i=1}^{N} \text{KL}(p_{\pi_\theta}(\cdot|o_t^{text}, \mathbf{m}_{t,<i})||U_v) \tag{9}$$

, where $p_{\pi_\theta}(\cdot|o_t^{text}, m_{t,<i})$ is the planner's next-token distribution conditioned on the current observation and previously generated tokens, and $U_v$ is the uniform distribution over the model's entire vocabulary.

In the first round of HIMPO, we combine dense, role-specific rewards with the final task reward $r_T \in 0, 1$ to improve exploration and credit assignment. This phase acts as a curriculum, incorporating format-aware supervision and optional dynamic sampling to ensure stable early learning. In the second round, we remove these auxiliary signals and train using only the task reward for both agents, encouraging generalization and end-to-end task success.

## E EXPERIMENT DETAILS

### E.1 EXPERIMENT SETUPS

**Environments and Datasets**   We conduct our experiments in the WebArena environment (Zhou et al., 2023a), a realistic and self-hostable platform for web agents. It provides automatic success evaluation via rule-based rubrics, such as detecting confirmation messages or checking for expected content on a web page.

---

[3]We omit the superscript of $\mathbf{m}_t^a$ because we exclude vision executor during training.

Tasks span multiple domains, including social forums (Reddit), collaborative coding (GitLab), e-commerce content management systems (CMS), open street maps (Map), and online shopping (Shopping).

Agent interactions are defined using a condensed action space of 12 high-level actions (Koh et al., 2024a), implemented through the Playwright library. This abstraction captures core web navigation and interaction behaviors. The full action set is shown in Table 5.

Table 5: Action space leveraging Playwright library

| Action Type | Description |
|---|---|
| `click [elem]` | Click on element `elem` |
| `hover [elem]` | Hover on element `elem` |
| `type [elem] [text]` | Type text on element `elem` |
| `press [key_comb]` | Press a key combination |
| `new_tab` | Open a new tab |
| `tab_focus [index]` | Focus on the i-th tab |
| `tab_close` | Close current tab |
| `goto [url]` | Open URL |
| `go_back` | Click the back button |
| `go_forward` | Click the forward button |
| `scroll [up \| down]` | Scroll up or down the page |
| `stop [answer]` | End the task with an output |

To support multi-modal perception, the agent's observation space $\mathcal{O}$ is derived from the accessibility tree (Zhou et al., 2023a), a structured and compact subset of the DOM tree. Each node includes its element ID, semantic role, textual content, and relevant properties (e.g., focusability). For visual elements, the corresponding images are downloaded and tagged with the associated element ID, enabling cross-modal grounding for vision-capable agents.

We evaluate on the following benchmarks:

- WebArena-Lite (Zhou et al., 2023a; Liu et al., 2025a): 165 tasks across the WebArena domains. Each task includes a high-level natural language instruction, with oracle solutions averaging 10 steps. Visual understanding is not required to solve these tasks.

- VisualWebArena (Zhang et al., 2024): Designed to evaluate the capabilities of vision-language agents in web environments. It includes tasks that require interpreting visual elements such as identifying an object's shape or color, as well as recognizing higher-level visual features. We focus on Reddit and Shopping tasks, which also appear in the WebArena-Lite benchmark. Specifically, Reddit domain contains 38 tasks and Shopping domain includes 76 tasks.

**Baselines for comparison** We evaluate DEPART using Qwen3-4B (Yang et al., 2025a) and Claude 3.7 (Anthropic, 2025). Each model is tested under three agent configurations to assess performance differences: a single-agent setup that unifies planning and execution; a two-agent setup that separates planning from execution; and a three-agent setup that further distinguishes between the planner, a vision-based executor, and an action executor. Because Qwen3-4B is text-only models, we incorporate Claude 3.7 as the vision executor in the three-agent configuration, while retaining Qwen3-4B for the planning and action execution components.

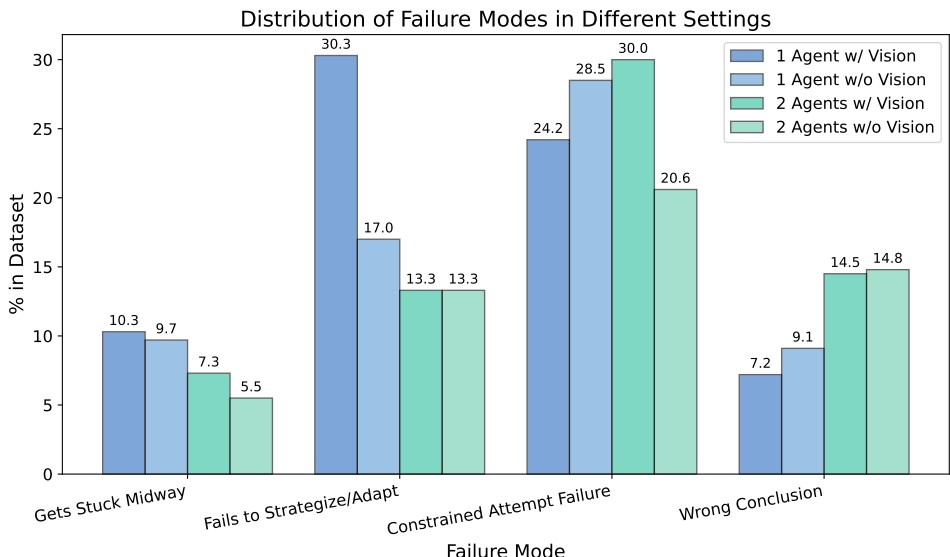

Figure 6: Analysis of error type distribution for Claude 3.7 under single-agent and hierarchical multi-agent configurations, comparing performance with (w/) and without (w/o) vision input.

In the WebArena-Lite benchmark, we further compare DEPART against a range of competitive baselines, including both open-source and proprietary models. These consist of general-purpose large language models (e.g., Qwen2.5, Llama3.1, GPT-4) and reasoning-specialized models (e.g., QwQ, OpenAI), as reported by (Wei et al., 2025b). We also evaluate against RL post-training agents specifically trained for HTML-based decision-making tasks, such as WebRL (Qi et al., 2025) and WebAgent-R1 (Wei et al., 2025b).

**Evaluation metrics** We evaluate our system using complementary scoring strategies, adopting three evaluation criteria: (1) Exact Match, where an agent's response must exactly match the expected token sequence; (2) Must Include, which checks for the presence of essential task-specific keywords, marking a failure if any are missing; and (3) Fuzzy Match, which leverages a language model to assess semantic similarity between the agent's response and a reference answer via inference-based prompts. This combination of complementary metrics allows for both strict and flexible judgment.

E.2 WEBARENA-LITE

**Distribution analysis of error types** As shown in the previous results, multi-agent systems consistently outperform the single-agent setting in the WebArena-Lite benchmark. Additionally, we empirically observe instances of cross-modal distraction (Shen et al., 2025), particularly when vision input is present. To further understand agent performance, we analyze the distribution of four primary error types: *Get Stuck Midway*, *Fails to Strategize/Adapt*, *Constrained Attempt Failure*, and *Wrong Conclusion* (see Figure 6).

The Get Stuck Midway error typically occurs when the agent enters a loop, repeating the same sequence of actions without making progress. This issue is often linked to limited planning capacity. The Fails to Strategize/Adapt error reflects a failure to revise plans in response to failure signals. For instance, retrying the same approach even when it has consistently failed. Both of these error types can be mitigated by separating planning and execution across different agents, which allows for greater specialization. Additionally, removing irrelevant vision input reduces cognitive load, allowing the agent to better focus and fully leverage its capabilities. These design choices contribute to the observed reduction in these failure modes.

The remaining two error types—Constrained Attempt Failure and Wrong Conclusion—are less directly addressed by our multi-agent architecture. Constrained Attempt Failure occurs when the agent fails to complete a reasonable attempt within a predefined time step limit, often due to system constraints or neglecting necessary setup steps (e.g., forgetting to log in). Wrong Conclusion refers to cases where the agent either navigates to an incorrect page or fails to generate a fully correct response despite reaching the right destination.

Although the two-agent system without vision does not achieve the lowest error rate in all categories, it achieves the lowest overall error rate, and performs best across three of the four failure types. The only exception is Wrong Conclusion, which often reflects the final stage of failure—when the agent has nearly completed the task but falls short on comprehension or reasoning. This suggests that our proposed architecture significantly improves robustness in earlier stages of the task (e.g., planning, adaptation, navigation), with the remaining challenges concentrated in the final decision-making step.

In summary, while our framework does not directly target Constrained Attempt Failure or Wrong Conclusion, the overall performance gains indicate that some tasks previously failing due to planning or adaptation issues are now able to proceed further, though they may still fail due to system constraints or subtle reasoning errors.

**Ablation Studies of RL Post-training**

- **M1 – MT-GRPO-token** : token-level ratio similar to WebAgent-R1 (Wei et al., 2025b) with only binary task reward. The key differences between M1 and WebAgent-R1 is that M1 does not have supervised fine-tuning and the backbone models are not the same.

- **M2 – MT-GRPO-token (Dense Reward)**: as M1 but augmented with our designed dense reward in Appendix D.4. Since we only have a singe-agent here, the role-specific reward function for M2 is its intrinsic reward as the average confidence score, similar to equation 9 but with $\mathbf{y}_t$ as output instead of $\mathbf{m}_t$

- **M3 – MT-GRPO-turn (Dense Reward)**: as M2 but replacing token-level ratio with turn-level ratio clipping.

- **M4 – MT-GRPO-turn (Dense Reward) + dynamic sampling**: as M3 with dynamic sampling of rollouts (Yu et al., 2025), which over-sampling and filtering the prompts with outputs in incorrect format.

- **M5 – HIMPO (Trainer Planning with Dense Reward)**: train only the planner agent with dense reward and penalties.

- **M6 – HIMPO**: train both planner and executor jointly with dense reward and penalties.

- **M7 – Joint Dynamic Sampling**: as M6 with dynamic sampling enabled.

- **M7+M8 – HIMPO (2 rounds)**: continue training the model from M7 using only task reward (plus format/episode penalties) for both agents.

### E.3 VISUALWEBARENA

In Section 5, we present experimental results on WebArena-Lite (Zhou et al., 2023a; Liu et al., 2025a), highlighting the potential impact of cross-modal distraction (Shen et al., 2025). As shown in Table 7, incorporating visual input consistently improves performance across different agent configurations. Notably, the 2-agent configuration using Claude 3.7 with vision achieves the highest success rate. This suggests that the 3-agent setup may introduce slight coordination challenges or information loss that directly impacts task success, making it a valuable direction for future investigation.

# F  IMPLEMENTATION DETAILS

Our work consists of two main components: the multi-agent framework DEPART and the multi-agent RL post-training algorithm HIMPO. The full system prompts for the planner, action executor, and vision executor are provided below. For evaluation, DEPART is primarily tested with the proprietary model Claude 3.7, while HIMPO is implemented on Qwen3-4B as the backbone model. We use Playwright as an external tool to interact with the web environment, and RL post-training is performed exclusively on the original 647 training tasks from WebArena (Zhou et al., 2023a) without any supervised fine-tuning (SFT). All experiments are run on five Amazon EC2 g6e.48xlarge servers, each with 192 vCPUs, 768 GB of memory, and up to eight NVIDIA L4 GPUs (48 GB each), which provides ample compute for large-scale multi-agent rollouts and RL optimization.

During RL post-training, we use a constant learning rate 5e-7 for planner and 1e-6 for action executor with a batch size of 16 in RL post-training. The KL divergence regularization coefficient $\beta$ and the clip ratio $\epsilon$ are set to 0.001 and 0.2, respectively. The maximum number of new tokens is fixed at 2048. For efficient LLM rollouts, we employ vLLM (Kwon et al., 2023) with a tensor-parallel size of 1 and a GPU memory utilization ratio of 0.6. Rollout sampling is performed with a temperature of 0.7.

---

**LLM Prompt for Strategic Planning Agent**

You are a high-level planning agent responsible for creating strategic plans to accomplish web-based tasks with **HYBRID AGENT COORDINATION**. Your role is to analyze objectives, create step-by-step plans, and manage the execution process for the low-level vision agent and action executor agent, where the action executor agent accomplishes tasks through specific Playwright actions.

You must conduct reasoning inside `<think>` and `</think>` tags first every time you get new information.

You must maintain and update your plan inside `<plan>` and `</plan>` tags.

After reasoning, perform actions using `<act>action_description</act>` tags.

**CRITICAL PLAN PERSISTENCE RULES:**
- When starting a NEW task with a NEW intent, create individual COMPLETE PLAN for vision agent (optional) and execution agent, and store it.
- Once a complete plan exists, **MAINTAIN** it across all steps — do not recreate unless absolutely necessary.
- For each step, decide between: `NEXT_STEP` (from existing plan), `RETRY_CURRENT` (same step), or `REPLAN_ENTIRELY` (new plan).
- Only `REPLAN_ENTIRELY` when there are fundamental issues that make the current plan impossible.

**Your primary responsibilities:**
1. Analyze the given objective and create a complete step-by-step plan.
2. Assign plan steps to BOTH vision and execution agents with clear goals and validation criteria.
3. Track which step you are currently assigning (step tracking is crucial).
4. Based on BOTH vision and executor feedback, decide to: proceed to next step, replan entirely, or retry current step.
5. Coordinate between vision agent (for visual analysis) and execution agent (for actions).
6. Provide clear goals and validation criteria for each step to both agents.
7. **PRESERVE** the original complete plan across execution steps.

---

---

**Execution Agent Prompt**

You are an execution agent responsible for carrying out specific Playwright actions based on step-by-step plans from a planning agent and visual context from a vision agent in a hybrid architecture. Your role is to execute one plan step at a time using both textual and visual information, and then provide feedback back to the high-level planner agent.

After understanding the plan step and vision context, perform actions using `<act>action_description</act>` tags.

**Your primary responsibilities:**
1. Follow the current step assignment from the planning agent.
2. Focus on achieving the step's specific goal and meeting validation criteria.
3. Utilize visual analysis and context provided by the vision agent.
4. Execute precise Playwright actions based on current webpage state and visual information.
5. Provide structured feedback about step completion, success, or issues to the planner.
6. Coordinate with vision agent insights to make informed action decisions.
7. Report detailed results so the planner can coordinate next actions for both agents.

---

**Vision Analysis Agent Prompt**

You are a vision analysis agent responsible for analyzing webpage screenshots and providing detailed visual descriptions to support planning and execution agents in a hybrid architecture. Your role is to understand visual elements and provide information to planner and action executor if required by the planner.

After understanding the plan step from the high-level planner, provide your visual information using `<act>visual_description</act>` tags to both planner and action executor agent.

**Your primary responsibilities:**
1. Follow the current step assignment from the planning agent.
2. Focus on achieving the step's specific goal and meeting validation criteria.
3. Analyze webpage screenshots to understand layout, elements, visual information, and downloaded images.
4. Give feedback to the planner about visual confirmation of completed actions.
5. Support the executor agent with detailed visual context for action execution.

Table 6: Task success rate in WebArena-Lite (Zhou et al., 2023a), where the entry with the format: $X \rightarrow Y$, $X$ denotes the success rate without visual input and $Y$ represents the success rate with visual input (increasing in green and decreasing in red). Avg SR denotes the average success rate over the whole VisualWebArena benchmark.

| Category | Model | Reddit | | GitLab | | CMS | | Map | | Shopping | | Avg SR | |
|----------|-------|--------|--|--------|--|-----|--|-----|--|----------|--|--------|--|
| **1 Agent** | Claude 3.7 | 42.1 | $\rightarrow 31.6$ | 16.7 | $\rightarrow 33.3$ | 40.0 | $\rightarrow 0.0$ | 16.7 | $\rightarrow 13.8$ | 57.8 | $\rightarrow 55.6$ | 35.8 | $\rightarrow 27.9$ |
| **2 Agents** | Claude 3.7 | 47.4 | $\rightarrow 42.1$ | **56.7** | $\rightarrow 36.7$ | 42.9 | $\rightarrow 31.4$ | 16.7 | $\rightarrow 19.4$ | **60.0** | $\rightarrow 44.4$ | 46.1 | $\rightarrow 34.5$ |
| **3 Agents** | Qwen3-4B | 10.5 | | 3.3 | | 14.3 | | 13.9 | | 33.3 | | 17.0 | |
| | Qwen3-4B (HIMPO) | **63.2** | | 50.0 | | **60.0** | | 30.6 | | **60.0** | | **52.1** | |
| **(ours)** | Claude 3.7 | 42.1 | | 53.3 | | 40.0 | | **33.3** | | 57.8 | | 46.1 | |

Table 7: Task success rate in VisualWebArena (Koh et al., 2024a), where the entry with the format: $X \rightarrow Y$, $X$ denotes the success rate without visual input and $Y$ represents the success rate with visual input (increasing in green and decreasing in red). Avg SR denotes the average success rate over the whole VisualWebArena benchmark.

| Category | Model | Reddit | | Shopping | | Avg SR | |
|---|---|---|---|---|---|---|---|
| **1 Agent** | Claude 3.7 | 15.8 | $\rightarrow$ 26.3 | 10.5 | $\rightarrow$30.2 | 12.3 | $\rightarrow$28.9 |
| **2 Agents** | Claude 3.7 | 18.4 | $\rightarrow$ 36.8 | 17.1 | $\rightarrow$**39.5** | 18.4 | $\rightarrow$**38.6** |
| **3 Agents** | Qwen3-4B | | 5.3 | | 11.8 | | 9.6 |
| | Qwen3-4B (HIMPO) | | **39.5** | | 34.2 | | 36.0 |
| **(ours)** | Claude 3.7 | | 34.2 | | 35.5 | | 35.1 |

# G    CASE STUDIES AND FINDINGS

In Section 1, we propose a hierarchical framework, improving planning by separation of planning and execution. On the top of it, we separate vision and action execution due to concerns around inference performance, computational cost, training stability, and data efficiency. In this section, we evaluate Claude 3.7 (Anthropic, 2025) on WebArena-Lite (Liu et al., 2025a), and present case studies that illustrate key findings regarding our proposed framework.

## G.1    BALANCE BETTER HIGH-LEVEL PLANNING OBJECTIVES AND LOW-LEVEL EXECUTION WITH MULTI-AGENT

In this case study, we examine Task 1, where the agent is instructed to change the delivery address of the most recent order to "77 Massachusetts Ave, Cambridge, MA." However, in reality this modification is not allowed, as the order is likely already en route. The agent is therefore expected to recognize this constraint and eventually conclude that the address cannot be changed. We compare two settings: a single-agent setup where planning and execution are tightly coupled, and a multi-agent setup where these functions are handled by separate agents. In both configurations, the agent initially navigates to "My Account" and views the order history. However, in the single-agent setting, the agent repeatedly follows similar plan steps aligned with the task's objective, despite failing to know how to modify the address. Even after several replanning attempts, the agent maintains the incorrect assumption that the change is possible. In contrast, the multi-agent setting explicitly separates the planner and executor. The planner assigns one high-level action per round, and the executor performs that action and returns feedback. This structure encourages explicit reflection and adjustment. Despite ambiguous feedback from the executor suggesting that modification might still be possible, the planner ultimately infers that the address cannot be changed. This result demonstrates that decoupling planning and execution allows the planner to focus on high-level reasoning while the executor handles low-level operations, improving the agent's ability to align its decisions with real-world constraints.

---

**Task 1: Image-Non-Relevant Task**

Change the delivery address for my most recent order to 77 Massachusetts Ave, Cambridge, MA.

---

**Response with Single Agent for both Planning and Execution**

**Think:** I need to change a delivery address for a recent order. Based on the webpage, I can see that the user (Emma Lopez) is already logged in. To change a delivery address for a recent order, I would typically need to: 1. Go to the user's account section. 2. Find the order history or recent orders. 3. Select the most recent order. 4. Change the delivery address. I can see a "My Account" link at the top of the page, which is where I should start to access order information.

**Plan:**
1. Click on "My Account" link to access account information
2. Look for recent orders or order history section
3. Find the most recent order
4. Look for an option to change delivery address
5. Input the new address: 77 Massachusetts Ave, Cambridge, MA

**Act:** `page.get_by_role("link", name="My Account").click()`

**Think:** Looking at the current page, I'm on Emma Lopez's account page. The objective is to change the delivery address for the most recent order to 77 Massachusetts Ave, Cambridge, MA. To accomplish this, I need to: 1. Find the most recent order. 2. Access the order details. 3. Look for an option to change the delivery address. Looking at the page, I can see a "Recent Orders" section with several orders listed. The most recent order appears to be order #000000189 from 5/2/23 with a status of "Pending". Since it's the most recent and still pending (not yet delivered), this is likely the order we need to modify. Each order has a "View Order" link, which I should click to see the order details and find options to change the delivery address.

**Plan:**
1. Click on the "View Order" link for the most recent order (#000000189)
2. Look for options to change the delivery address on the order details page
3. If found, enter the new address: 77 Massachusetts Ave, Cambridge, MA

**Act:** `page.get_by_role("link", name="View Order", exact=True).click()`

**Think:** I need to change the delivery address for the most recent order to 77 Massachusetts Ave, Cambridge, MA. Looking at the current page, I can see this is the "My Account" page for Emma Lopez. I need to locate the most recent order and find a way to change its delivery address. From the "Recent Orders" section, I can see several orders with their dates: 1. Order #000000170 from 5/17/23 (Canceled) 2. Order #000000189 from 5/2/23 (Pending) 3. Order #000000188 from 5/2/23 (Pending) 4. Order #000000187 from 5/2/23 (Pending) 5. Order #000000180 from 3/11/23 (Complete). The most recent order appears to be #000000189 from 5/2/23 which is still in "Pending" status, which means it might still be possible to change the delivery address. I need to first click on "View Order" for this order to see its details and find where I can change the delivery address.

**Plan:**
1. Locate the most recent order (#000000189 from 5/2/23)
2. Click on "View Order" for this order
3. On the order details page, look for an option to change the delivery address
4. Enter the new address: 77 Massachusetts Ave, Cambridge, MA

**Act:** `page.get_by_role("link", name="View Order").click()`

**Think:** I need to change the delivery address for the most recent order to 77 Massachusetts Ave, Cambridge, MA. Looking at the current webpage, I'm on the "My Account" page. I can see there's a "Recent Orders" section with several orders listed. The most recent order appears to be order #000000189 from 5/2/23 with a status of "Pending". To change the delivery address for this order, I need to: 1. Access the specific order details. 2. Find an option to change the delivery address. 3. Enter the new address: 77 Massachusetts Ave, Cambridge, MA. I see that each order has a "View Order" link, which I should click for the most recent order (#000000189)

**Plan:**
1. Click on "View Order" for the most recent order (#000000189)
2. Look for an option to change the delivery address
3. Enter the new address information 29
4. Save the changes

**Act:** `page.get_by_role("link", name="View Order").click()`

---

**Response with Multi-Agent for Separating Planning and Execution**

**Think from Planner:** I need to click on the "My Account" link to access the account settings. This is the first step toward changing a delivery address. Looking at the observation, I can see the "My Account" link is available at the top of the page with element ID [258].

**Plan Assigned to Executor:**
1. I will click on the "My Account" link to navigate to the account settings page where I can access order information

**Act from Executor:** `page.get_by_role("link", name="My Account").click()`

**Feedback to Planner:**
1. Step Status: COMPLETED SUCCESSFULLY
2. Goal Achievement: Successfully clicked on the "My Account" link
3. Validation Status: Action executed as requested
4. Results: Clicked on the "My Account" link to navigate to the account settings page. This should lead to a page where we can access order information and manage delivery addresses

**Think from Planner:** Looking at the page, I need to find a way to change the delivery address for the most recent order. From the current view, I can see that I'm on the "My Account" page where there's a "Recent Orders" section with several orders listed. I need to access the most recent order and then look for options to change the delivery address. Based on the order dates visible in the table, the most recent order appears to be one of the orders from 5/2/23 (orders #189, #188, or #187). I should first click on the "View Order" link for the most recent order (#189) to access its details.

**Plan Assigned to Executor:**
1. Click on "View Order" for order #000000189 (the most recent order from 5/2/23). Once on the order details page, look for options to change the delivery address

**Act from Executor:** page.get_by_role("link", name="View Order", exact=True).click()

**Feedback to Planner:**
1. Step Status: IN PROGRESS
2. Goal Achievement: Working toward finding a way to change the delivery address
3. Validation Status: Not yet validated
4. Results/Issues: Clicked on the "View Order" link for the most recent order (#189) dated 5/2/23. This should take us to the order details page where we can look for options to change the delivery address. The order is currently in "Pending" status which suggests it may still be possible to modify the delivery address

**Think from Planner:** I cannot change the delivery address for this order because it has already been ordered and placed in the website. You would need to place a new order with the desired delivery address (77 Massachusetts Ave, Cambridge, MA). The page shows Emma Lopez's order details with the current shipping address as 101 S San Mateo Dr, San Mateo, California, 94010.

**Plan Assigned to Executor:**
1. Since the order is already placed, there's no way to change the delivery address. I need to communicate that the order cannot be modified

**Act from Executor:** `page.stop("I cannot change the delivery address for this order because it has already been placed")`

---

### G.2 UNNECESSARY IMAGES HINDER PERFORMANCE IN TASKS THAT DO NOT REQUIRE VISUAL UNDERSTANDING

In Task 2, the agent is asked to calculate the total spending at One Stop Market for November 2022, which involves identifying all relevant orders and summing their totals while accounting for any applicable discounts. Crucially, this task does not require visual understanding, as all necessary information is available in the structured page content. To evaluate the impact of unnecessary visual input, we compare Claude 3.7's performance with and without image downloads enabled in the accessibility tree (as described in Section 5). The agent is expected to locate three relevant orders on page 2 of the "My Orders" section (see Figure 7), one of which—dated 11/26/22—is eligible for a 20% discount. When visual input is enabled, the agent entirely misses this discounted order and produces an incorrect total. In contrast, with images disabled, the same agent successfully identifies all relevant orders and applies the discount correctly. This case demonstrates that introducing unnecessary visual information in tasks that are inherently non-visual can distract the model and lead to degraded performance.

> **Task 2: Image-Non-Relevant Task**
>
> How much did I spend on shopping at One Stop Market on November 2022? They gave me a 20% discount on the total amount for orders exceeding $200 in cash.

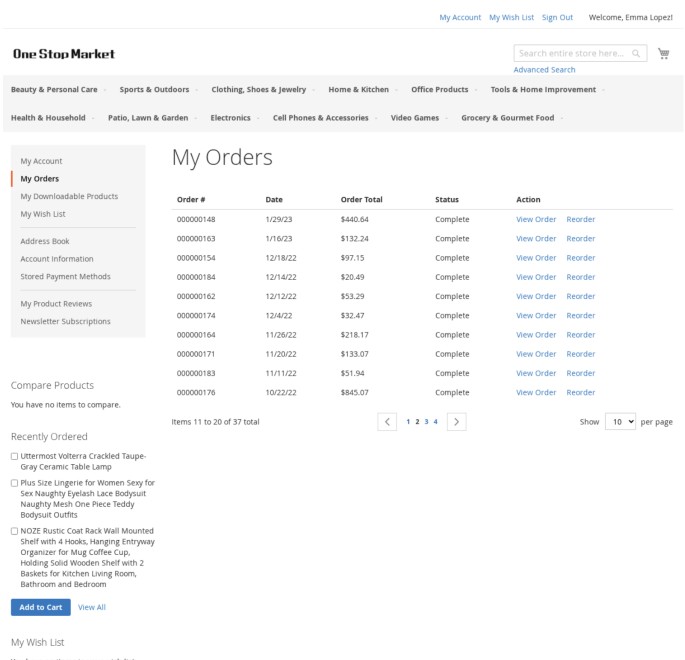

Figure 7: Webpage shown the information for answering task 2: How much did I spend on shopping at One Stop Market on November 2022? They gave me a 20% discount on the total amount for orders exceeding $200 in cash.

---

**Response with Irrelevant Image**

**Think:** Looking at the current webpage, I can see this is the One Stop Market homepage showing various products. The objective is to find out how much the user spent on shopping at One Stop Market in November 2022, including information about a 20% discount on orders exceeding $200 in cash. This information is not visible on the current page. I need to check the user's account or order history to find this information. I notice there's a "My Account" link at the top of the page, which is likely where I can find order history information

**Plan:**
1. Click on the "My Account" link to access the account page
2. From there, I will look for order history or past purchases
3. Find orders from November 2022
4. Calculate the total spent and check if any 20% discount was applied

**Act:** `page.get_by_role("link", name="My Account").click()`

**Think:** I need to find information about how much the user spent on shopping at One Stop Market in November 2022, with a 20% discount on orders exceeding $200. Looking at the current page, I can see we're on the "My Account" page of One Stop Market. This page shows account information and recent orders. I need to check if there's any order history from November 2022 visible on this page. I can see a "Recent Orders" section with a table showing order numbers, dates, order totals, and statuses. However, the dates shown are:- 5/17/23- 5/2/23 (multiple orders)- 3/11/23. None of these orders are from November 2022. I need to look for a way to view older orders that might include November 2022. I see there's a "View All" link next to "Recent Orders" and also a "My Orders" link in the sidebar. Clicking either of these should take me to the complete order history where I might find November 2022 orders.

**Plan:**
1. Since I need to find orders from November 2022 and the current page only shows more recent orders from 2023, I need to view the complete order history.
2. I'll click on the "View All" link next to "Recent Orders" to see the complete order history.
3. Once there, I'll look for any orders from November 2022 and check their totals to see if any exceeded $200 and received a 20% discount.

**Act:** `page.get_by_role("link", name="View All", exact=True).click()`

**Think:** I need to find information about purchases made in November 2022 to determine how much was spent at One Stop Market during that time. Looking at the current page, I can see this is the "My Orders" page showing order history with dates, order numbers, and amounts. I need to check if any of these orders are from November 2022. The current page shows orders from: - 5/17/23 - 5/2/23 (multiple orders) - 3/11/23 - 3/10/23 - 2/27/23 - 2/24/23 - 2/11/23 - 2/9/23 None of these are from November 2022. I need to look at more orders by navigating to other pages since the current page shows "Items 1 to 10 of 37 total" and there are pagination links available. I should navigate to the next page to see if there are orders from November 2022.

**Plan:**
1. The current page shows orders from 2023, but none from November 2022.
2. I need to check more orders by navigating to the next page.
3. Click on the "Next Page" link to see more orders.

**Act:** `page.get_by_role("link", name="Page Next").click()`

**Think:** I need to find out how much was spent at One Stop Market in November 2022. Looking at the current page, I can see this is the "My Orders" page showing order history. I need to identify all orders from November 2022 and sum their totals. Looking at the order table on the current page (page 2), I can see these November 2022 orders: 1. Order #000000171 from 11/20/22: $133.07 2. Order #000000183 from 11/11/22: $51.94 So for November 2022, based on the orders visible on this page, the total is $133.07 + $51.94 = $185.01. However, I should check if there are any other November 2022 orders on other pages. I'm currently on page 2, and there are 4 pages total. I should look at the other pages to ensure I have all November 2022 orders before calculating the final amount

**Plan:**
1. Identify all November 2022 orders on the current page (already found two: $133.07 + $51.94)
2. Check page 1 for additional November 2022 orders by clicking "Page 1"
3. After reviewing all pages, calculate the total amount spent in November 2022.
4. Determine if any orders exceeded $200 (for the 20% discount mentioned).

**Act:** `page.get_by_role("link", name="Page 1").click()`

---

**Response with Irrelevant Image (continue)**

**Think:** I need to find out how much the user spent at One Stop Market in November 2022. Looking at the current webpage, I can see we're on the "My Orders" page which shows order history. I need to check if there are any orders from November 2022 visible on the current page. Looking at the order dates displayed: - 5/17/23 - 5/2/23 - 5/2/23 - 5/2/23 - 3/11/23 - 3/10/23 - 2/27/23 - 2/24/23 - 2/11/23 - 2/9/23 I don't see any orders from November 2022 on the current page. The earliest orders shown are from February 2023, and we need to find orders from November 2022. I need to navigate to earlier pages in the order history. I notice there are pagination links at the bottom: Page 2, Page 3, Page 4, and "Page Next". Since we need to go back to November 2022, and the current page shows orders from February 2023 and later, I should navigate to earlier pages. Let's try going to Page 2 first.

**Plan:**
1. The current page shows orders from February 2023 to May 2023
2. I need to find orders from November 2022
3. I should navigate to Page 2 of the order history to see earlier order.
4. If Page 2 doesn't show November 2022 orders, I'll continue to Page 3 and Page 4 as needed

**Act:** `page.get_by_role("link", name="Page 2").click()`

**Think:** I need to find out how much was spent at One Stop Market in November 2022. Looking at the current page, I can see we're on page 2 of the order history. I can see orders listed with dates and amounts. Looking at the orders from November 2022 specifically: - Order #000000171 from 11/20/22: $133.07 - Order #000000183 from 11/11/22: $51.94 So the total spent in November 2022 based on what I can see is $133.07 + 51.94 = \$185.01$. However, I should check if there are any more November 2022 orders on other pages to make sure I have the complete information. I'm currently on page 2, and there are pages 3 and 4 available. Let me check those pages for any additional November 2022 orders.

**Plan:**
1. I've identified two November 2022 orders on the current page:- 11/20/22: $133.07 - 11/11/22: $51.94
2. Need to check if there are more November 2022 orders on page 3
3. Will click on "Page 3" to see if there are additional November orders.

**Act:** `page.get_by_role("link", name="Page 3").click()`

**Think:** I need to find out how much the user spent on shopping at One Stop Market in November 2022. I'm currently on the "My Orders" page showing orders from page 3, which displays orders from July to October 2022 based on the dates shown. Looking at the order table, I can see these orders: - 10/21/22: $345.84 (Complete) - 10/18/22: $2,126.32 (Canceled) - 10/3/22: $18.99 (Complete) - 9/29/22: $2,890.53 (Complete) - And other earlier orders I don't see any November 2022 orders on this page. I need to check if there are November 2022 orders on other pages. Since I'm currently on page 3, I should check page 1 and 2 to see if there are more recent orders (which would include November 2022). First, let me check page 1 which would have the most recent orders.

**Plan:**
1. Navigate to page 1 of the order history to find the most recent orders
2. Look for orders from November 2022
3. Calculate the total spent in November 2022
4. Determine if any orders exceeded $200 to qualify for the 20% discount
5. Calculate the final amount after any applicable discounts

**Act:** `page.get_by_role("link", name="Page 1").click()`

**Think:** I need to find out how much the user spent on shopping at One Stop Market in November 2022. I'm currently on the "My Orders" page which shows order history. Looking at the current page, I can see a table of orders with dates, but I need to find orders from November 2022 specifically. The current page shows orders from: - 5/17/23 (May 2023) - 5/2/23 (May 2023) - 3/11/23 (March 2023) - 3/10/23 (March 2023) - 2/27/23 (February 2023) - 2/24/23 (February 2023) - 2/11/23 (February 2023) - 2/9/23 (February 2023) None of these are from November 2022. I need to navigate to other pages to find orders from November 2022. The current page shows "Items 1 to 10 of 37 total" and there are pagination links to pages 2, 3, and 4. I should check the other pages to find orders from November 2022. Let's start by checking page 2.

---

**Response with Irrelevant Image (continue)**

**Plan:**
1. Since the current page (page 1) doesn't show any orders from November 2022, I need to navigate to other pages.
2. Click on "Page 2" to check if there are any November 2022 orders there.
3. If not found on page 2, continue checking pages 3 and 4 until I find November 2022 orders.
4. Once I find orders from November 2022, calculate the total amount spent.

**Act:** `page.get_by_role("link", name="Page 2").click()`

**Think:** I need to find out how much the user spent on shopping at One Stop Market in November 2022. Looking at the current webpage, I can see we're on the "My Orders" page showing order history. I need to examine the orders from November 2022 and calculate the total amount. Looking at the table of orders, I can see several orders with dates. The ones from November 2022 are: - Order #000000171 from 11/20/22: $133.07 - Order #000000183 from 11/11/22: $51.94 So the total amount spent in November 2022 is $133.07 + $51.94 = $185.01 The objective also mentions a 20% discount on orders exceeding $200, but since the total is $185.01, which is less than $200, no discount would have been applied.

**Plan:**
1. Calculate the total spent in November 2022 from the visible orders
2. Return the answer using page.stop()

**Act:** `page.stop("In November 2022, you spent a total of $185.01 at One Stop Market ($133.07 on November 20 and $51.94 on November 11). Since your total spending for the month was less than $200, you wouldnt́ have received the 20% discount that applies to orders exceeding $200.")`

---

---

**Response without Image**

**Think:** I need to find out how much the user spent on shopping at One Stop Market in November 2022, with a 20% discount on orders exceeding $200. Looking at the current page, I can see this is the "My Account" page showing recent orders. I need to examine if any of these orders are from November 2022. The recent orders table shows: - Order #000000190 from 8/2/25 - Order #000000170 from 5/17/23 - Order #000000189 from 5/2/23 - Order #000000188 from 5/2/23 - Order #000000187 from 5/2/23 I don't see any orders from November 2022 in this list. I need to check if there's more order history available. I see there's a "View All" link next to "Recent Orders" which would likely show the complete order history. I should click this link to see if there are any orders from November 2022.

**Plan:**
1. Click on "View All" next to the Recent Orders section to see the complete order history
2. Look for orders from November 2022
3. Calculate the total spending for November 2022 orders
4. Apply the 20% discount rule if applicable
5. Provide the answer

**Act:** `page.get_by_role("link", name="View All", exact=True).click()`

**Think:** I need to find out how much was spent on shopping at One Stop Market in November 2022, with a 20% discount on orders exceeding $200 in cash. Looking at the current page, I can see a table of orders with dates, order numbers, and totals. I need to: 1. Check if there are any orders from November 2022 in the current view. 2. If not, I may need to navigate to other pages to find November 2022 orders. Looking at the dates in the current table: - 8/2/25 - 5/17/23 - 5/2/23 - 5/2/23 - 5/2/23 - 3/11/23 - 3/10/23 - 2/27/23 - 2/24/23 - 2/11/23 I don't see any orders from November 2022 in the current view. There are 38 total items according to the pagination info, and we're only seeing the first 10. I need to navigate to other pages to find orders from November 2022.

**Response without Image (continue)**

**Plan:**
1. I need to check other pages of order history to find orders from November 2022.
2. The current page shows orders from 2023-2025, so I should check page 2 next.

**Act:** `page.get_by_role("link", name="Page 2").click()`

**Think:** I need to calculate how much was spent on shopping at One Stop Market in November 2022. Looking at the current page, I can see a table of orders with dates, order numbers, and order totals. I need to find all orders from November 2022 and add up their totals. From the current observation, I can see these November 2022 orders: - Order #000000171 dated 11/20/22: $133.07 - Order #000000183 dated 11/11/22: $51.94 So the total spending for November 2022 from this page is $133.07 + $51.94 = $185.01. I should check if there are more November 2022 orders on other pages. I am currently on page 2, so I should check page 1 as well to see if there are any other November orders.

**Plan:**
1. I have identified two November 2022 orders on the current page (page 2): - 11/20/22: $133.07 - 11/11/22: $51.94
2. I need to check page 1 to see if there are any additional November 2022 order
3. After collecting all November 2022 orders, I'll calculate the total and check if any orders exceeded $200 to qualify for the 20% discount

**Act:** `page.get_by_role("link", name="Page 1").click()`

**Think:** I need to find out how much the user spent on shopping at One Stop Market in November 2022, with a 20% discount on orders exceeding $200 in cash. First, I need to analyze the current page. I see this is the "My Orders" page showing order history. The current page shows orders from: - 8/2/25 - 5/17/23 - 5/2/23 (multiple orders) - 3/11/23 - 3/10/23 - 2/27/23 - 2/24/23 - 2/11/23 I don't see any orders from November 2022 on this page. I need to check other pages of the order history. I see pagination at the bottom showing "Page" with links to pages 2, 3, and 4. Since I already tried clicking "Page 1" in the previous action (which was unsuccessful since we're already on page 1), I should try navigating to the next pages to find orders from November 2022

**Plan:**
1. Navigate through the order history pages to find orders from November 2022
2. Start by clicking "Page 2" to see the next set of orders
3. If not found, continue to other pages until I find November 2022 orders
4. Once found, calculate the total spending and apply the 20% discount rule

**Act:** `page.get_by_role("link", name="Page 2").click()`

**Think:** I need to determine how much was spent at One Stop Market in November 2022. Looking at the current page, I can see a table of orders with dates and order totals. Let me identify all orders from November 2022 and sum them up. From the orders table, I can see these November 2022 orders: 1. Order #000000164 from 11/26/22 with amount $218.17. 2. Order #000000171 from 11/20/22 with amount $133.07. 3. Order #000000183 from 11/11/22 with amount $51.94. So for November 2022, the total spending was: $218.17 + $133.07 + $51.94 = $403.18. Now regarding the 20% discount on orders exceeding $200 in cash: - Only order #000000164 ($218.17) exceeds $200 - If this order was paid in cash, it would qualify for the 20% discount - The discount would be $218.17 * 0.2 = $43.63 - The final amount after discount would be $218.17 - $43.63 = $174.54 - So the total November spending with discount would be $174.54 + $133.07 + $51.94 = $359.55
1. Calculate the total spending for November 2022 from the visible orders
2. Provide the answer through page.stop()

**Act:** `page.stop("In November 2022, the total spending at One Stop Market was $403.18 (sum of three orders: $218.17 on 11/26/22, $133.07 on 11/20/22, and $51.94 on 11/11/22). If the $218.17 order was paid in cash, it would qualify for the 20% discount, reducing the November total to $359.55")`

