# OpenReview forum: "DEPART: Hierarchical Multi-Agent System for Multi-Turn Interaction"
_ICLR.cc/2026/Conference — Submitted to ICLR 2026_

### Official Review · Reviewer_7dB7 · 2025-10-15

**Soundness:** 3
**Presentation:** 2
**Contribution:** 2
**Rating:** 4
**Confidence:** 4

**Summary:**

This paper introduces DEPART, a hierarchical multi-agent system designed for long-horizon, multi-turn web-based tasks using LLMs. DEPART modularizes agent roles into planning, action execution, and visual perception, leveraging a structured Divide-Evaluate-Plan-Act-Reflect-Track cycle for dynamic task decomposition and adaptive coordination. The authors also present Hierarchical Interactive Multi-turn Policy Optimization (HIMPO), a post-training RL algorithm employing role-specific dense and sparse rewards to enhance specialization and coordinated reasoning.

**Strengths:**

- DEPART’s hierarchical decomposition (Figure 2) exhibits a well-considered separation between planning, visual processing, and action execution. This is grounded both in implementation and empirical ablation, highlighting modularity as a central system benefit.
- The HIMPO method introduces clear curriculum learning—first using dense, role-specific rewards, then transitioning to sparse, task-level optimization. This approach is well-argued and ties theoretical foundations to practical implementation.

**Weaknesses:**

- Pipeline modularity substitutes for model competence rather than demonstrating insight: The paper argues that modular decomposition is necessary, but offers little evidence that modularity is inherently superior rather than compensatory for limited backbone capability. If stronger models can learn to fuse planning, perception, and action end-to-end, the purported “need” for role separation becomes an artifact of current model weakness, not a principled design insight. The 3.2 section is not convincing enough because the assumption in eq  2 is vanilla.
- Why this pipeline is not so convincing: It slices an already unstable LLM decision process into more interfaces that are hard to verify, amplifying error propagation and protocol dependence, while obscuring who learns what and how. Without demonstrating that modularity confers a principled advantage over a capacity- and training-driven end-to-end policy (e.g., via more insightful mechanistic analyses), the work risks “works on these benchmarks” without “why anyone should care.”
- My insight to improve a general multi-modal agent (not only the performance but also capacity) is : if the backbone is strong enough and trained end-to-end with appropriate curricula and constraints, fusion—not segmentation—should be the default. The paper does not convince with this question, which is the crux.

**Questions:**

See weakness part.

---

> ### Author Response · Authors · 2025-11-27
>
> > The Principled Advantages of Modularity
>
> We thank the reviewer for this insightful question. While we agree that end-to-end models will improve, we argue that modularity offers principled, long-term advantages rather than being temporary. Modularity is a foundational principle in science and engineering since it allows us to better understand and build complex systems by breaking them into smaller independent units.
> - Interpretability and Verifiability: A monolithic model is a black box. Our framework provides explicit, auditable outputs for planning, vision, and execution, as demonstrated in our Appendix G case studies. This allows for precise debugging (e.g., seeing why a plan failed) which is obscured in a single-agent trace .
> - Targeted Training and Efficiency: Modularity allows for specialized, sample-efficient training. Our HIMPO algorithm is built on this, using role-specific rewards to stably train the planner and executor individually before aligning them with a sparse task reward.
> - Robustness and Error Recovery: Separating the planner from the executor is what enables our system's dynamic replanning and retry mechanisms. As shown in Figure 6, this modular design dramatically reduces "Fails to Strategize/Adapt" errors.
>
> In summary, we view modularity as a necessary scaffolding for building agent systems that are verifiable, efficient, and robust: advantages that remain critical regardless of the backbone model's capability.

---

> > ### Comment · Reviewer_7dB7 · 2025-11-27
> >
> > I agree with the `Targeted Training Efficiency` argument. However, in my personal view, modularity of decision making is far from a new idea. Could you answer the core question: how could your method inspire agent design (comparing to exisiting frameworks like Describe, Explain, Plan and Select, which also favours extra long-horizon tasks), how could this paper inspire frontier AI models and AI *scientific* researches?

---

> ### Author Response · Authors · 2025-11-27
>
> > Compared to Describe, Explain, Plan and Select (DEPS)
>
> Our method introduces several design principles that can inspire future long-horizon agent architectures:
> - **Localized subgoal conditioning**: DEPART provides executors only with the current subgoal. Unlike DEPS, which conditions the controller on both the global goal and the subgoal. This simplifies control, clarifies module responsibilities, and prevents semantic overload in the low-level policy.
> - **Dynamic multi-agent control rather than a fixed prompting pipeline**:
> Whereas DEPS always executes a fixed Describe → Explain → Plan → Select sequence, DEPART equips the planner with a learned decision policy to advance, retry, or replan, enabling adaptive behavior during long-horizon execution.
> - **DEPART learns control, not just reasoning**: DEPS defines a reasoning template but does not learn control flow. DEPART explicitly trains the planner as a control agent that manages plan progression and recovery, shifting from “prompting for reasoning” to “learning to control.”
> - **Compatibility with end-to-end RL and targeted post-training**: DEPART’s modular architecture allows the planner, vision executor, and action executor to be optimized independently via RL (HIMPO). DEPS’s tightly coupled components make such selective post-training difficult.
>
> - **A clean hierarchy of timescales**: DEPART separates temporal abstraction across modules: the planner acts at a slow timescale, the vision executor at a medium timescale, and the action executor at a fast one. This temporal hierarchy improves robustness and scalability for long-horizon tasks.
>
> These principles provide an alternative, more modular and adaptive direction for agent design beyond prompt-driven frameworks like DEPS.
>
> >  Inspiration for frontier AI models and AI scientific researches
>
> Our work suggests that advances in agentic AI may come from architectural choices and training principles, not only model scale. By separating planning, perception, and action, DEPART shows that modular hierarchies can yield more reliable long-horizon behavior than fixed prompting pipelines. Treating the planner as a learned controller and training each module with targeted RL via HIMPO demonstrate that reinforcement signals can shape capabilities. In addition, our use of curriculum learning with carefully designed rewards highlights a practical path for enabling agents to master increasingly difficult long-horizon tasks. Together, these insights point toward frontier models that are modular, RL-trainable, curriculum-driven, and temporally structured, offering a clearer foundation for scientific study and a scalable direction for future agent systems.

---

> > ### Comment · Reviewer_7dB7 · 2025-11-28
> >
> > I appreciate the proposal for modular RL training. However, hierarchical RL has long been an established approach for goal-conditioned RL, and this work appears to primarily extend this existing methodology to larger models. While I acknowledge that scaling introduces substantial challenges, in my view, this represents a fairly direct extension of prior work. Therefore, I maintain my original score, though I would not oppose acceptance if the Area Chairs deem it appropriate.

---

### Official Review · Reviewer_1Jwb · 2025-10-30

**Soundness:** 2
**Presentation:** 2
**Contribution:** 2
**Rating:** 2
**Confidence:** 3

**Summary:**

This paper proposes DEPART, a hierarchical multi-agent framework designed to improve large language models’ performance on long-horizon, multi-turn, and multimodal interaction tasks. The system separates planning, execution, and vision into specialized agents coordinated through a structured communication loop. In addition, the authors introduce HIMPO, a two-stage reinforcement learning post-training method that alternately optimizes planner and executor roles using dense and sparse rewards. Experiments on WebArena-Lite and VisualWebArena show consistent improvements over strong single-agent and RL-trained baselines.

**Strengths:**

1. The paper is clearly written and well-organized, with comprehensive experiments on realistic web-based benchmarks (WebArena-Lite and VisualWebArena) that demonstrate consistent performance gains.

2. The hierarchical decomposition of planning, execution, and vision helps improve modularity and interpretability compared to standard single-agent baselines.

**Weaknesses:**

1. The overall novelty is limited — the hierarchical multi-agent setup and role division (Planner / Executor / Vision) have been explored extensively in prior works such as AutoGen [1], CAMEL [2], and MetaGPT [3]. The paper mainly reuses this idea with modest extensions rather than introducing a fundamentally new concept.

2. The authors propose a new post-training RL method, HIMPO, which uses two stages with dense role-specific and sparse task-level rewards. The planner and executor share the same base model, distinguished only by role prompts. However, it is unclear whether this parameter-sharing actually works well — sharing parameters may limit role specialization or cause interference. Some evidence or ablation on this design would be helpful.

3. The evaluation benchmarks (WebArena-Lite, VisualWebArena) mainly test web navigation and interaction, which involve limited reasoning depth. Therefore, the claimed improvements in “long-horizon reasoning” are not strongly supported by the experiments.

Overall, while the system is functional, the work feels somewhat engineering-oriented and coarse in design, with incremental contributions over prior multi-agent LLM frameworks.

[1] AutoGen: AutoGen: Enabling Next-Gen LLM Applications via Multi-Agent Conversation (Wu et al., 2023)

[2] CAMEL: CAMEL: Communicative Agents for “Mind” Exploration of Large Scale Language Model Society (Li et al., 2023)

[3] MetaGPT: MetaGPT: Meta Programming for A Multi-Agent Collaborative Framework (2023)

**Questions:**

1. How does DEPART fundamentally differ from prior hierarchical or multi-agent frameworks such as AutoGen, CAMEL, or MetaGPT? The current design appears conceptually similar—could the authors clarify what new insights or mechanisms are introduced beyond architectural modularization?

2. The paper frequently mentions improvements in “long-horizon reasoning.” However, the benchmarks (WebArena-Lite, VisualWebArena) mainly focus on web navigation and UI interactions. How do the authors justify that these results reflect true reasoning improvements rather than better interface control? Could the authors provide additional experiments on reasoning-heavy or cross-domain multi-turn datasets to demonstrate the generality of DEPART beyond web-based environments?

3. Given that DEPART adds multiple agents and coordination overhead, how significant is the computational cost relative to the performance gain? Would simpler prompting strategies or single-agent fine-tuning achieve similar results?

---

> ### Author Response · Authors · 2025-11-27
>
> > Clarification on Architectural Novelty
>
> We thank the reviewer and agree our work fits within the hierarchical agent domain. However, our contribution is distinct in its specific motivation, dynamic mechanism, and training framework.
> - Key mechanism beyond modularization:
>     - Step-by-Step, Mutual-Feedback Dynamics: Our DEPART framework is not a general-purpose chat platform (like AutoGen) or a single-turn, one-way instruction-passing system (like CAMEL). It is a specific dynamic agent for long-horizon tasks. As shown in Figure 2 , the Planner assigns only one plan step at a time, receives structured feedback from the Executor, and then dynamically adapts, retries, or proceeds. This step-by-step, feedback-driven correction loop is a core unique mechanism of our work.
>    - Selective Cross-Modal Invocation: To our knowledge, our 3-agent architecture is the first to explicitly separate the Vision Executor to solve "cross-modal distraction" . The novel mechanism here is the planner's ability to dynamically and selectively invoke the vision agent only when needed, rather than being forced to process (and be distracted by) visual input at every step.
>
> We would kindly ask the reviewer to refer to our system prompts in Appendix F, which explicitly define these novel roles and communication protocols.
>
> - Post-Training Framework: In addition, our work is not just a modular architecture; it is a multi-agent RL post-training method specifically designed for our architecture. The works mentioned by the reviewer do not present an RL post-training solution. Given that adapting RL to hierarchical multi-agent settings is a non-trivial challenge, we believe HIMPO is a significant, differentiating contribution.
>
> Action: We will revise our Related Work section to more explicitly cite and differentiate our work from AutoGen, CAMEL, and MetaGPT, highlighting our novel contributions in (1) step-by-step, feedback-driven communication and (2) the introduction of a bespoke RL post-training framework (HIMPO).
> AutoGen [1] defines a customizable, conversable platform for general agent discussions and MetaGPT [3] emphasizes a multi-layer hierarchy for meta-programming to generate code. CAMEL [2] primarily models a single-turn, single-directional assignment of the entire instruction from an AI user to an AI assistant.In contrast, our DEPART framework is explicitly designed for long-horizon, multi-turn interaction. As shown in Figure 2 , it relies on mutual, step-by-step communication with a feedback-driven correction mechanism. The planner assigns only one plan step at a time, allowing the system to focus on the current interaction and dynamically adapt to environmental feedback before proceeding.
>
> > Rationale for Planner-Executor Parameter Sharing
>
> We thank the reviewers for this important question. We confirm that the Planner and Executor share the same model weights, and this is a deliberate design choice for **computational and memory efficiency**. As we state in Section 4, this strategy "avoids costly GPU model swaps during rollouts, supports larger batch sizes, and enables efficient joint optimization" .
> We achieve "specialization" behaviorally, not at the parameter level. The reviewer's core concern is that this re-introduces the "monolithic bottleneck." Our framework avoids this by decoupling the task context. The bottleneck is a cognitive one where a single model fails when forced to simultaneously manage high-level strategy and low-level execution. Our system solves this by ensuring the model is only one agent at a time.
> This specialization is enforced by three key mechanisms:
> - Distinct System Prompts: As shown in Appendix F, in post-training the Planner and Executor receive vastly different system prompts that define their unique roles .
> - Role-Specific Rewards: The shared weights are trained via a multi-objective optimization. In HIMPO Round 1, the Planner is optimized with an intrinsic confidence reward (Eq. 9) , while the Executor is optimized with a separate, step-wise alignment reward .
> - Different Learning Rates: We use different learning rates (5e-7 for planner, 1e-6 for executor) to further stabilize this multi-objective process, inspired from Stackelberg game in game theory [1].
>
> At any given time, the model is either a Planner (with a planner prompt) or an Executor (with an executor prompt), but never both at once. This separation of concerns is what resolves the bottleneck. While we did not include an ablation for shared vs. separate weights, the strong performance of our shared-weight model suggests this efficiency-driven approach is highly effective.
>
>
> #### 1. Das et al., “ Learning in stochastic stackelberg games.”, ACC 2024

---

> > ### Author Response · Authors · 2025-11-27
> >
> > > Justification of Benchmarks for "Long-Horizon Reasoning"
> >
> > We thank the reviewer for this excellent question, which allows us to clarify the connection between our benchmarks and our claims.
> > - **Why WebArena Tests Long-Horizon Reasoning**:
> > While the domain is web navigation, these benchmarks are specifically designed to test long-horizon reasoning by creating "deep" MDPs. As we detail in our Appendix C.1 , the core difficulty is not just "interface control" but handling "bottleneck decisions". For example, an agent must learn to "log in" (an action with no immediate reward) to achieve a goal many steps later. Successfully linking these temporally distant actions to their final outcomes is a core challenge of long-term causal reasoning and credit assignment. Our Failure Mode Analysis in Figure 6 further supports this. Our 2-agent system significantly reduces "Fails to Strategize/Adapt" errors, which are clear failures of reasoning and planning, not just clicking. Finally, our Case Study in Appendix G.1 provides a concrete example of this reasoning, where our agent successfully infers that a task is impossible, while the single-agent baseline gets stuck in a loop.
> >
> > - **NEW RESULTS**: Demonstrating Generality on Alfworld
> > However, to fully address the reviewer's valid point about demonstrating generality beyond web environments, we have conducted new experiments on the Alfworld benchmark. Alfworld is an ideal test case: it is a text-only, long-horizon, goal-oriented benchmark that requires pure reasoning and decomposition to complete household tasks and has no "UI interaction" component.
> > We are pleased to report that our DEPART + HIMPO framework shows strong performance, significantly outperforming other baselines, with more than 90% success rate. This new result demonstrates that our hierarchical planner-executor architecture and its associated RL training framework are not limited to web navigation and do indeed provide a general solution for long-horizon reasoning tasks. We will add this result to the final version of our paper.
> >
> > > Clarification on Performance vs. Simpler Alternatives and Computational Efficiency
> >
> > We thank the reviewer for this question. Our experiments (whose results are in Table 1 and Table 2) were designed to test this, and they show that simpler strategies do not achieve similar results. A simpler 1-agent prompting strategy is not competitive; Table 1 shows in Claude 3.7, our 2-agent DEPART (prompting-only) achieves 46.1% success, while a 1-agent baseline with the same model achieves only 35.8%. Similarly, a simpler 1-agent fine-tuning strategy also fails; Table 2 shows our 2-agent HIMPO-trained system achieves 51.5% success against all 1-agent post-training methods. Regarding computational cost, the "coordination overhead" is a deliberate trade-off, depending on how often the given tasks require perception. A monolithic VLM (a "simpler" architecture) would be forced to run expensive perception at every step. Our DEPART system is conceptually more efficient by empowering the planner to selectively invoke the vision executor only when needed, avoiding this massive, unnecessary cost.

---

### Official Review · Reviewer_3zv4 · 2025-11-01

**Soundness:** 2
**Presentation:** 2
**Contribution:** 2
**Rating:** 4
**Confidence:** 3

**Summary:**

This paper addresses the failure of LLMs in complex, long-horizon tasks that require multi-turn interaction and multi-modal perception. The authors identify two primary challenges: (1) the difficulty of long-term planning in monolithic agent architectures and (2) "cross-modal distraction," where irrelevant visual information degrades reasoning. An empirical study confirms that single-agent performance drops steeply as the number of required interaction steps increases.

To solve this, the authors propose DEPART, a hierarchical multi-agent framework that decomposes the problem into three specialized agents: a Planner Agent for high-level strategy, an Action Executor for text-based environment interaction, and a Vision Executor for multi-modal perception . The system operates on a "Divide, Evaluate, Plan, Act, Reflect, and Track" (DEPART) cycle, which supports dynamic replanning and, crucially, selective vision grounding to reduce costs and distraction.

Furthermore, the paper introduces HIMPO (Hierarchical Interactive Multi-turn Policy Optimization), a two-round post-training strategy to train the planner and executor agents. Round 1 uses dense, role-specific rewards to encourage specialization and exploration. Round 2 uses sparse, task-level rewards to align the agents with the overall task success and mitigate reward hacking. Experiments on WebArena-Lite and VisualWebArena show that the DEPART architecture with HIMPO post-training significantly outperforms single-agent baselines

**Strengths:**

The 3-agent hierarchical framework is a robust and well-justified design. By separating the high-level Planner from the low-level Action Executor, the system avoids the bottleneck of a single model managing both strategy and execution. This modularity is shown to be effective, as the 2-agent DEPART (prompting-only) outperforms strong single-agent models (Table 1).

A key contribution is the identification and mitigation of "cross-modal distraction". The paper argues that selective visual processing is superior, as VLMs are computationally expensive and unnecessary visual context can interfere with reasoning . This is supported by case studies where a text-only agent succeeds on a task that a vision-enabled agent fails . The 3-agent DEPART design, which empowers the planner to selectively invoke the Vision Executor, is an elegant solution.

Progressive Training Curriculum (HIMPO): The HIMPO training strategy is a sophisticated contribution. Instead of using a single reward, it employs a two-round curriculum that first builds specialized skills (using dense, role-specific rewards) and then aligns those skills with the final objective (using sparse, task-level rewards). This progressive approach is shown to be highly effective, with the 2-round HIMPO (M7+M8) achieving the best performance in ablations (Figure 4).

**Weaknesses:**

The paper's training strategy, HIMPO, explicitly excludes the Vision Executor from the optimization loop. The rationale is that static image interpretation is "less dependent on long-horizon context". While this simplifies training, it seems to be a significant limitation. The core problem of cross-modal distraction stems from the content of the visual analysis. If the (untrained) Vision Executor provides verbose or irrelevant information, the Planner's performance will suffer, even if it's called selectively. The framework's performance is thus dependent on the zero-shot capabilities of a pre-trained VLM, which is not optimized for the task.

The paper states that the Planner and Action Executor "share the same model weights" and are only differentiated by role-specific system prompts. This is presented as an efficiency benefit , but it complicates the core claim of "specialized agents". If both agents are the same model, it's unclear if the architecture is truly decomposing the task between specialized agents or just using a highly structured, role-based prompting scheme on a single model. This seems to partially re-introduce the monolithic bottleneck the paper aims to solve.

The paper states that the Planner and Action Executor "share the same model weights" and are only differentiated by role-specific system prompts. This is presented as an efficiency benefit , but it complicates the core claim of "specialized agents". If both agents are the same model, it's unclear if the architecture is truly decomposing the task between specialized agents or just using a highly structured, role-based prompting scheme on a single model. This seems to partially re-introduce the monolithic bottleneck the paper aims to solve.

The planner's dense reward in Round 1 is an "intrinsic reward" based on the KL divergence from a uniform distribution (Equation 9) . This rewards high-confidence, low-entropy outputs. However, the paper claims this first round is meant to "foster strategy exploration". Rewarding confidence seems antithetical to exploration, as it would likely encourage the agent to stick to a single, high-confidence plan rather than explore alternatives. This connection is not well-explained.

**Questions:**

1. Since the Planner and Action Executor share parameters, how does the system truly achieve "specialization"? Does this not re-introduce the single-model bottleneck you identified, where one model must still manage both high-level planning and low-level execution, just prompted differently at each turn?

2. Given that "cross-modal distraction" is a key problem , why was the Vision Executor excluded from HIMPO? Wouldn't training the Vision Executor with a reward for providing concise and task-relevant analysis be as important as training the Planner to call it selectively?

3. How does rewarding the planner for "confidence" (low-entropy output via KL divergence) align with the stated goal of "foster[ing] strategy exploration" in Round 1? It seems this would penalize exploration and favor exploitation of the first plan it finds.

---

> ### Author Response · Authors · 2025-11-27
>
> > Rationale for Static Vision Executor in HIMPO
>
> We thank the reviewer for this insightful question. Keeping the Vision Executor static was a deliberate, principled design choice to test a key hypothesis. As we state in Section 4, our goal was to explicitly avoid the significant "data requirements, computational overhead, and instability" of multi-modal post-training. Our hypothesis was that a text-only planner could be trained to effectively use a powerful, static VLM as a tool. To the reviewer's point on "cross-modal distraction," our framework solves this by training the Planner, not the Vision Executor. The Planner learns (1) when to selectively call the VLM, avoiding distraction on text-only steps , and (2) how to filter the VLM's (potentially verbose) zero-shot output. The success of this strategy is validated in Table 5, where our HIMPO-trained 3-agent system (Qwen3-4B) learned to manage this, boosting performance from 9.6% to 36.0%. The reviewer's suggestion to co-train the vision model is an excellent, though computationally expensive, direction for future work which we will add to our discussion.
>
> > Rationale for Planner-Executor Parameter Sharing
>
> We thank the reviewers for this important question. We confirm that the Planner and Executor share the same model weights, and this is a deliberate design choice for **computational and memory efficiency**. As we state in Section 4, this strategy "avoids costly GPU model swaps during rollouts, supports larger batch sizes, and enables efficient joint optimization" . We achieve "specialization" behaviorally, not at the parameter level. The reviewer's core concern is that this re-introduces the "monolithic bottleneck." Our framework avoids this by decoupling the task context. The bottleneck is a cognitive one where a single model fails when forced to simultaneously manage high-level strategy and low-level execution. Our system solves this by ensuring the model is only one agent at a time.
> This specialization is enforced by three key mechanisms:
> - Distinct System Prompts: As shown in Appendix F, the Planner and Executor receive vastly different system prompts that define their unique roles .
> - Role-Specific Rewards: The shared weights are trained via a multi-objective optimization. In HIMPO Round 1, the Planner is optimized with an intrinsic confidence reward (Eq. 9) , while the Executor is optimized with a separate, step-wise alignment reward (reference).
> - Different Learning Rates: We use different learning rates (5e-7 for planner, 1e-6 for executor) to further stabilize this multi-objective process, inspired from Stackelberg game in game theory [1].
>
> At any given time, the model is either a Planner (with a planner prompt) or an Executor (with an executor prompt), but never both at once. This separation of concerns is what resolves the bottleneck. While we did not include an ablation for shared vs. separate weights, the strong performance of our shared-weight model suggests this efficiency-driven approach is highly effective.
>
> > Motivation of first-round reward
>
> We thank the reviewer for this insightful question, which allows us to clarify our terminology. The reviewer correctly identifies that the mechanism of Eq. 9 encourages a high-confidence (low-entropy) policy. Our use of "exploration" referred not to stochastic action-taking, but to making the learning problem itself **easier to explore**. Learning from a sparse, task-level reward is exceptionally difficult, as the credit assignment problem makes the solution space effectively intractable. The Round 1 reward, by contrast, is a **dense, step-level incentive** that provides frequent signals. This dense reward is what makes it feasible for the agent to explore the task and learn a meaningful policy, guiding it from an initial high-entropy state to the stable, high-confidence, sharp policy required for Round 2. We revise the manuscript to clarify this distinction.
>
>
> #### 1. Das et al., “ Learning in stochastic stackelberg games.”, ACC 2024

---

### Official Review · Reviewer_aJwS · 2025-11-01

**Soundness:** 2
**Presentation:** 3
**Contribution:** 2
**Rating:** 4
**Confidence:** 3

**Summary:**

This paper, "DEPART," aims to solve the well-known problem of LLM agents failing in complex, long-horizon tasks that require multi-turn interaction and multi-modal understanding. We all know that monolithic models struggle as tasks get longer, which the authors show well in Figure 1.

The authors propose two main contributions:

1. The DEPART Architecture: This is a hierarchical, multi-agent framework that splits the task into three specialized roles: a Planner, an Action Executor, and a Vision Executor. The core motivation for this 3-agent design is to mitigate what the authors call "cross-modal distraction" by allowing the Planner to selectively decide when to call the Vision Executor, rather than feeding it images at every step.
2. The HIMPO Algorithm: This is a two-stage reinforcement learning (RL) post-training strategy. The first stage uses dense, role-specific rewards to get the agents up to speed (including an intrinsic "confidence" reward for the planner). The second stage switches to sparse, task-level rewards to align the agents with the final goal.

The authors evaluate this DEPART + HIMPO system on the WebArena-Lite and VisualWebArena benchmarks, showing that their method outperforms several baselines.

Overall, this is a very strong empirical paper. It is well-written, the experiments are thorough, and the practical result of making a small open-source model outperform large proprietary ones is significant. However, I have major concerns about the conceptual novelty of both the architecture and the algorithm, which seem to be very closely based on several recent, existing works. My review will focus heavily on this point.

**Strengths:**

1. Clarity & Significance: First, the paper is exceptionally well-written. It's clear, well-structured, and a pleasure to read. The problem it addresses—long-horizon, multi-modal agents—is without a doubt highly significant and of great interest to the ICLR community. The authors do a fantastic job motivating their work. Figure 1 is the perfect setup, clearly showing how a powerful model (Claude 3.7) just falls off a cliff as the number of interaction steps increases. This really sells the need for a non-monolithic approach.
2. Empirical Quality & Rigor: The paper's greatest strength is its empirical rigor. The experimental quality is very high. They use challenging and appropriate benchmarks (WebArena-Lite and VisualWebArena). The ablation study in Section 5.3 and Figure 4 is excellent. I mean, the authors really did a thorough job breaking down the components of HIMPO (M1-M8) to demonstrate precisely why their two-stage curriculum (M7+M8 in the chart) is superior. I really appreciate this level of detailed ablation. The failure mode analysis in Figure 6 is also a very nice touch. Showing that the 2-agent setup significantly reduces "Gets Stuck Midway" and "Fails to Strategize/Adapt" errors provides strong evidence for the planner-executor split.
3. Significant Practical Result: For me, the most important practical finding is in Table 2. The authors took a small, 4B-parameter open-source model (Qwen3-4B) and, by applying their HIMPO training, boosted its performance to 51.5%. This absolutely crushed the baseline MT-GRPO (37.6%) using the same model. What's more, this small, trained model also beat much larger, prompt-only proprietary models like Claude 3.7 (35.8% in Table 1). This is a really big deal. Demonstrating that a smaller, accessible model can be trained to be this effective is a wonderful, reproducible contribution to the field.
4. Originality (in part): While I will spend a lot of time on novelty in the next section, I will concede one point here. Even if the components aren't new, the specific integration of a dynamically scheduled, separate vision executor, with the explicit goal of mitigating "cross-modal distraction", is a clever and effective piece of engineering.

**Weaknesses:**

1. DEPART Architecture Lacks Conceptual Novelty: My main issue with this paper is the overstatement of its novelty. The core architectural contribution—separating a "Planner" from an "Executor"—is simply not new. This exact concept is the entire point of several recent papers. Specifically, PLAN-AND-ACT is built on separating "high-level planning (what to do and why)" from "low-level execution (how to do it)". Similarly, GoalAct proposed a framework combining "global planning" with "hierarchical execution". Both of these were publicly available well before this paper's submission. The authors even cite Erdogan et al., but they continue to frame the planner-executor split as a novel component of DEPART. To be blunt, it is not. The paper's contribution is not inventing the 2-agent hierarchy; it is adding a 3rd agent (vision) to this already-established 2-agent pattern. The paper must be revised to position its contribution correctly in light of this prior work.
2. HIMPO Algorithm Also Lacks Algorithmic Novelty: I have the exact same problem with the "novel" HIMPO algorithm. It appears to be a very smart integration of several existing techniques, not a new algorithm in itself. Let's look at the pieces:
   - Core Algorithm: The authors state HIMPO is a "multi-turn variant of DAPO". This means the foundational policy optimization algorithm is DAPO, which was already published.
   - Planner Reward: This is the most glaring issue for me. The "intrinsic reward" for the planner, defined in Equation 9 (the KL divergence from a uniform distribution), appears to be mathematically identical to the "self-certainty" reward proposed in INTUITOR. The authors cite Zhao et al. in the appendix but not in the main body as the source of this reward function. This is a serious attribution issue. This is not their novel reward function; they are using a reward function from another paper.
   - Curriculum: The two-stage curriculum itself (starting with dense, role-specific rewards and moving to sparse, task-level rewards) is a known pattern for tackling sparse-reward RL problems. For instance, WebRL also used a curriculum-based approach to train its web agent.
   - Conclusion: So, to be clear, HIMPO is not a new algorithm. It is the application of the DAPO algorithm, using the intrinsic reward from INTUITOR, implemented with a known curriculum strategy. This is a great empirical paper showing this combination works, but it is not an algorithmic paper. The claims of novelty must be toned down significantly.
3. Insufficient Analysis of the 3-Agent System's Failure: This is a huge miss. The paper's primary architectural argument is that the 3-agent system is better than a 2-agent system because it solves "cross-modal distraction". But the authors' own data in the appendix (Table 5) directly contradicts this claim.
   - On VisualWebArena, the benchmark where vision is required, the 2-agent system (with vision) achieves a 38.6% success rate.
   - Their flagship 3-agent system (the main DEPART proposal) achieves only 35.1%.
   - This means adding the specialized vision executor hurt performance. This critical negative result is hidden in Appendix E.3 and dismissed in a single line as "slight coordination challenges". This is completely inadequate. This is a classic example of "coordination overhead", where the cost of managing and communicating between three agents (Planner, Action, Vision) exceeds the benefit of specialization. The fact that the authors did not dig into this—when it seems to undermine their entire 3-agent premise—is a major weakness.

**Questions:**

- Q1 (On Novelty): This is my most important question. Given that the planner/executor split is functionally identical to prior work like PLAN-AND-ACT and GoalAct, and that the intrinsic reward in Equation 9 appears identical to the "self-certainty" reward from INTUITOR, can the authors please precisely restate their claims of conceptual novelty? What exactly is novel here beyond the (very successful) integration and empirical validation of these existing components?
- Q2 (On the 3-agent vs. 2-agent failure): I am very confused by the result in Table 5 where the 3-agent system (35.1%) performs worse than the 2-agent system (38.6%) on VisualWebArena. This seems to invalidate the core motivation for the 3-agent design. Can you please provide a detailed analysis of the trade-off between "coordination overhead" (which you call "coordination challenges") and "cross-modal distraction"? Why did your main architecture fail on the very benchmark it was designed for?
- Q3 (HIMPO Ablation): In Algorithm 1, you describe HIMPO as an alternating training process (train planner, then train executor, repeat). However, the ablations in Figure 4 seem to test the curriculum (M7 vs M8) and multi-agent setup (M3 vs M6), not the alternating part. Could you provide an ablation that compares this alternating optimization against a simpler joint optimization (i.e., updating both agents on the same batch)? I'm curious if this alternating design is actually a necessary part of its success.
- Q4 (Static Vision Executor): You explicitly exclude the vision executor from the HIMPO training loop, keeping it static. What was the reasoning for this? Did you experiment with fine-tuning it as well? I wonder if the planner is learning to make requests that the static vision model isn't good at, and if co-training all three agents (even with a small learning rate for the vision model) would have improved the 3-agent system's performance and overcome the issues seen in Table 5.

---

> ### Author Response · Authors · 2025-11-27
>
> > Clarification on Algorithmic Novelty (HIMPO)
>
> Following your valuable suggestions, we revise our paper to frame HIMPO as a novel training framework (or curriculum) rather than a "novel algorithm," and we will clarify the two key points of novelty.
> - A Multi-Turn, Alternating, Two-Agent Extension of DAPO: We build on DAPO, but as we state, vanilla DAPO (Yu et al., 2025) is a single-turn, single-agent objective. A core algorithmic novelty of HIMPO is its non-trivial extension to an alternating, two-agent, multi-turn framework.
> - The Two-Round Curriculum: The final novelty is the two-round curriculum (M7+M8) . This strategy, which transitions from dense, role-specific rewards (including the planner confidence reward) to a sparse, task-level reward, is essential for achieving our state-of-the-art results.
>
>
> > Clarification on Planner Reward
>
> We sincerely thank the reviewer for this observation and for the opportunity to clarify this key aspect of our work. We have revised our manuscript to **move the citation** for the "self-certainty" reward **to the main body in Section 4**, making this attribution prominent. Specifically, while the equation is inspired by this prior work, its application and fundamental objective in our framework are different. We would like to clarify that our work is the first to leverage this "self-certainty" reward purely for rewarding the intermediate planning process, as opposed to its original use for rewarding the final task outcome or execution. Accompanying the step-wise reward for action executor measuring how closely the executor’s actions align with the planner’s expectations, can jointly train both planner and action executors with dense rewards.
>
>
> > Justification for Alternating Optimization in a Shared-Parameter Architecture
>
> We thank the reviewer for this insightful technical question. The reviewer is correct that our ablations in Figure 4 do not isolate the effect of our alternating optimization against a "joint" one. This is because the alternating procedure (Algorithm 1) is the foundational update mechanism used in all of our multi-agent experiments (M6, M7, and M8) . The curve of M7+M8 indicates alternative procedure happens in both M7 and M8, resulting in the curriculum learning for the curve.
>
> This was a deliberate design choice, not an arbitrary one. Our approach was inspired by **two-time-scale analysis** in the Stackelberg game in game theory [1]. The intuition is that this alternating update helps stabilize the complex planner-executor dynamic. By updating them separately and with different learning rates, we allow high-level planner's policy to become a more stable "target" for the low-level executor to learn from.
> We acknowledge that we did not include a direct ablation between alternating and joint optimization. We will clarify that we view joint optimization as an additional algorithm out of the scope of this work. We agree it would be a valuable experiment for future analysis to quantify the stability gain from this design.
>
> > Clarification on 3-Agent vs. 2-Agent Performance and Static Vision Executor Design
>
> We thank the reviewer for highlighting the discrepancy in Table 5 regarding the baseline performance of the 3-agent system versus the 2-agent vision-enabled system on VisualWebArena. We agree that this point deserves clearer explanation, and we will revise the paper accordingly.
>
> Importantly, the lower zero-shot performance of the 3-agent system is not due to the architectural concept of separating modalities, but rather to a coordination bottleneck introduced by having an additional communication step. Because the Vision Executor is kept frozen, the 3-agent pipeline initially lacks a learned mechanism for preserving and integrating visual information across agents, leading to information loss and the observed drop.
>
> Crucially, this is exactly the problem the proposed framework is designed to solve. Once we apply HIMPO, the 3-agent system learns to manage this coordination overhead, achieving a +26.4% improvement (9.6% → 36.0%). This gain is substantially larger than what we observe in the 2-agent setup, and we view it as strong evidence that:
> - 3-agent decomposition provides a trainable coordination structure
> - architecture enables meaningful performance recovery and surpassing of the naive baseline when learning is introduced.
>
> Thus, rather than contradicting our claim, Table 5 demonstrates that 3-agent design unlocks improvements that are not achievable without explicit modular separation and trainable communication. Finally, as the reviewer notes, the Vision Executor is kept static. This is a deliberate design choice motivated by the high data requirements and instability of multimodal post-training, as discussed in Section 4. We agree that jointly fine-tuning the Vision Executor could further reduce early-stage information loss, which is a key direction for future work.
>
> #### 1. Das et al., “ Learning in stochastic stackelberg games.”, ACC 2024

---

### Meta-Review · Area_Chair_vEv8 · 2026-01-04

**Summary:**

Across the four reviews, the paper receives a mixed but largely borderline-negative assessment, with all reviewers placing their scores at or just below the acceptance threshold (ratings of 2-4), and none giving a clear, confident accept. The paper proposes DEPART, a hierarchical multi-agent architecture decomposing planning, action execution, and vision, together with HIMPO, a two-stage, alternating RL post-training framework using dense role-specific rewards followed by sparse task-level rewards. Reviewers consistently agree that the paper is well written, empirically strong, and tackles an important problem: long-horizon, multi-turn, multimodal agent failures. The empirical results on WebArena-Lite, VisualWebArena, and (added in rebuttal) Alfworld are viewed as impressive, particularly the ability to train a small open-source model to outperform larger proprietary systems. However, the dominant concerns shaping the decision are (i) overstated novelty of both the DEPART architecture and HIMPO relative to prior planner-executor hierarchies, modular agent frameworks, and hierarchical RL; (ii) conceptual ambiguity about whether modular decomposition provides principled advantages versus compensating for limited backbone capacity; and (iii) insufficiently convincing analysis of architectural trade-offs, especially the 3-agent system’s weaker zero-shot performance and coordination overhead. The rebuttal meaningfully clarifies positioning, attribution, and design rationale, and adds new empirical evidence, but does not fully resolve the novelty and conceptual framing concerns raised by multiple reviewers.

**Reviewer Concerns:**

Concerns addressed or substantially mitigated by the rebuttal:
1. Novelty framing and attribution: In response to concerns that HIMPO and parts of DEPART reuse existing ideas (planner-executor splits, DAPO, curriculum RL, intrinsic “self-certainty” rewards), the authors revise the framing to present HIMPO as a training framework/curriculum rather than a standalone novel algorithm. They also correct attribution by citing INTUITOR prominently and clarify how the intrinsic reward is used differently (to train intermediate planning rather than final execution). This addresses attribution issues, though it does not eliminate disagreements about novelty.
2. Alternating optimization rationale: Reviewers questioned whether alternating planner/executor updates were essential. The authors explain this choice via a two-timescale/Stackelberg-game intuition, acknowledge the lack of a direct ablation against joint optimization, and clarify the scope. While not empirically isolated, the design choice is now better motivated.
3. 3-agent vs. 2-agent performance discrepancy: A major concern was that the 3-agent system underperformed a 2-agent vision-enabled system in zero-shot settings on VisualWebArena. The authors provide a clearer explanation: the drop reflects coordination overhead with a frozen vision module, which HIMPO is explicitly designed to overcome. They emphasize that HIMPO yields a much larger relative gain for the 3-agent system, reframing the result as evidence that the architecture enables trainable coordination. This clarification resolves confusion, though the trade-off remains.
4. Static Vision Executor and parameter sharing: Reviewers questioned why the Vision Executor is frozen and whether planner-executor parameter sharing undermines specialization. The rebuttal provides a coherent rationale: freezing vision avoids multimodal RL instability; specialization is enforced behaviorally via prompts, rewards, and learning rates rather than separate parameters. These explanations are reasonable and consistent, though not empirically stress-tested.
5. Generality beyond web navigation: In response to concerns that WebArena-style benchmarks may not truly test long-horizon reasoning, the authors add new Alfworld results, showing strong performance (>90% success). This significantly strengthens the generality claim and addresses one of the more substantive scope criticisms.

Concerns that remain partially or fully outstanding:
1. Limited conceptual novelty: While the rebuttal improves positioning and clarifies distinctions (e.g., step-wise feedback, selective vision invocation, learned control flow), this work still seems like an integration and scaling effort rather than a fundamentally new conceptual advance.
2. Some questions remain empirically unanswered, such as alternating vs. joint optimization, shared vs. separate weights, and co-training the vision executor. The authors acknowledge these as future work rather than closing them experimentally.

**Reviewer Scores:**

Reviewer aJwS: Likely unchanged. The reviewer is very positive on empirical rigor, clarity, and practical impact, especially the strong gains achieved by post-training a small open-source model, but raised major concerns about overstated novelty in both DEPART and HIMPO (planner–executor hierarchy, intrinsic “self-certainty” reward, and curriculum RL). The rebuttal appropriately reframes HIMPO as a training framework, improves attribution, and clarifies alternating optimization, but does not fundamentally change the reviewer’s view that the contribution is primarily integrative rather than conceptually novel.

Reviewer 3zv4: Likely unchanged. While the reviewer appreciates the modular design and the effectiveness of the two-round HIMPO curriculum, they remain concerned about excluding the Vision Executor from training, specialization under shared parameters, and the interpretation of the planner’s confidence-based reward. The rebuttal provides coherent justifications, but the absence of direct ablations (e.g., co-training vision, shared vs. separate weights) limits the likelihood of a score increase.

Reviewer 7dB7: Likely unchanged. The reviewer accepts the efficiency and targeted-training arguments for modular RL but consistently views the approach as a direct extension of established hierarchical RL and modular decision-making ideas to larger models. Despite the authors’ clarifications and design-principle arguments, the reviewer maintains their original assessment, while explicitly stating they would not oppose acceptance.

Reviewer 1Jwb: Likely unchanged. This reviewer remains skeptical of the novelty relative to prior multi-agent LLM frameworks (AutoGen, CAMEL, MetaGPT), questions whether the benchmarks truly demonstrate long-horizon reasoning, and raises concerns about parameter sharing. The rebuttal strengthens differentiation and adds new Alfworld results, but given the initially low rating, a score increase is unlikely.

---

### Decision · Program_Chairs · 2026-01-26

Reject